# DARE: Difficulty-Aware Dynamic Routing for Mixture of Experts

## Abstract

Sparse Mixture-of-Experts (MoE) architectures have become a foundational approach for efficiently scaling Large Vision-Language Models (LVLMs), as they activate only a subset of parameters for each input. However, the commonly adopted Top-K routing strategy assigns a fixed number of experts to every token, ignoring the natural variation in token complexity. This static allocation often results in suboptimal resource utilization, where simple tokens receive excessive computation and complex tokens are insufficiently processed. While recent dynamic routing methods attempt to address this limitation, they lack principled mechanisms to explicitly guide expert allocation based on token-level difficulty, resulting in suboptimal performance in practice. In this paper, we propose **D**ifficulty-**A**ware Dynamic **R**outing for Mixture of **E**xperts (**DARE**), a novel routing strategy that adapts expert selection according to the complexity of each token. DARE introduces a lightweight predictor that estimates the difficulty of individual tokens based on their log-perplexity as a theoretically grounded proxy, and employs a set of learnable thresholds to dynamically determine the appropriate number of experts to activate. This mechanism enables fine-grained and adaptive allocation of computational resources, allowing the model to devote more capacity to challenging tokens while conserving resources on easier ones. Extensive experiments on standard vision-language benchmarks demonstrate that DARE consistently outperforms both fixed Top-K routing and existing adaptive routing strategies. It achieves superior task performance while simultaneously improving computational efficiency, using 39% fewer experts compared to the Top-K baseline, highlighting the effectiveness and generality of difficulty-aware routing in sparse MoE architectures for large-scale multimodal models.

## 1 Introduction

Large Vision-Language Models (LVLMs) have achieved remarkable performance across a broad range of multimodal tasks, primarily driven by the continuous increase in model parameters and training data (Liu et al., 2023c; Wang et al., 2024b). While this scaling trend has led to significant advances in model capability, it also brings substantial computational costs, making large models increasingly difficult to train, deploy, and serve in real-world scenarios. Sparse Mixture-of-Experts (MoE) architectures (Shazeer et al., 2017) addresses these scaling challenges by activating only a small subset of specialized experts for each input, thereby substantially increasing model capacity without a corresponding rise in computational burden. Following their successful integration into Transformer-based architectures (Lepikhin et al., 2020), MoE techniques have been widely adopted in large-scale language, vision, and multimodal systems (Li et al., 2022; Dai et al., 2024; Lin et al., 2024; Wu et al., 2024a), demonstrating their scalability and versatility across domains.

A Mixture-of-Experts (MoE) layer consists of a set of expert subnetworks and a gating mechanism that assigns each input token to a selected subset of experts. The most widely adopted routing strategy, Top-K routing (Shazeer et al., 2017), directs each token to the top K experts with the highest gating scores. While simple and effective, this approach relies on a fixed allocation scheme that assumes uniform token complexity. In reality, the difficulty of tokens can vary considerably due to task-specific requirements (Rogers et al., 2021), lexical characteristics (Schick & Schütze, 2020), and contextual dependencies (Guu et al., 2020). This mismatch between variable token complexity and uniform expert allocation leads to systematic inefficiencies: computationally simple tokens

consume unnecessary expert capacity while complex tokens receive insufficient processing power. Beyond wasting computational resources, this imbalance limits the model's ability to adequately process challenging inputs, ultimately degrading overall performance.

While recent studies have identified the limitations of static expert allocation and developed a range of dynamic routing strategies by adjusting expert activation through reinforcement learning, heuristic rules, or the use of dummy experts (Huang et al., 2024; Guo et al., 2024; Jin et al., 2024; Zeng et al., 2024; Wang et al., 2024c; Yue et al., 2024; Lewis et al., 2021), most existing methods overlook a critical factor in routing decisions: the role of token-level difficulty. As shown in Figure 1, these approaches generally lack an explicit and interpretable mechanism for assessing token complexity and incorporating it into the routing process. This omission prevents models from making informed distinctions between computationally simple and complex tokens, resulting in continued resource misallocation despite the dynamic nature of these routing strategies.

To address these fundamental limitations, we propose **DARE** (**D**ifficulty-**A**ware Dynamic **R**outing for Mixture of **E**xperts), a principled routing mechanism that dynamically adapts expert assignment based on the difficulty of each input token. DARE introduces a lightweight predictor that estimates token complexity using log-perplexity–a theoretically grounded proxy corresponding to the cross-entropy loss that provides stable and interpretable difficulty signals. To determine how many experts should be activated, DARE employs an adaptive thresholding mechanism with learnable parameters that maps predicted difficulty scores to optimal expert counts within a predefined range. This design enables fine-grained, token-specific expert allocation, allowing the model to assign more computational capacity to harder tokens while minimizing redundancy for easier ones, creating a natural alignment between computational demand and expert utilization.

Our main contributions are summarized as follows:

- We identify a critical gap in existing dynamic routing strategies for Mixture of Experts, namely the absence of explicit guidance informed by token difficulty. To address this gap, we propose **DARE**, a novel routing strategy that dynamically adapts expert capacity based on a direct and interpretable measure of token complexity.

- We formally establish log-perplexity as a robust proxy for token difficulty and design a lightweight prediction module to estimate this difficulty at the token level. Leveraging a learnable threshold adaptation mechanism, DARE enables fine-grained expert selection by assigning a variable number of experts per token according to its estimated complexity.

- Through extensive experiments on standard vision-language benchmarks, we demonstrate that DARE consistently outperforms both traditional Top-K routing and recent adaptive routing methods. Our approach not only enhances task performance but also reduces computational cost by allocating resources more efficiently and effectively.

## 2 RELATED WORKS

**Large Vision-Language Models.** The success of large language models (LLMs) has spurred their extension to multimodal domains, leading to the emergence of large vision-language models (LVLMs). These models typically align visual and textual features via projection layers and benefit from scaling model size and pretraining data (Li et al., 2023b; Liu et al., 2023c; 2024b). However, such scaling incurs significant computational costs. To address this challenge, recent studies have introduced sparse Mixture-of-Experts (MoE) architectures into LVLMs through pretraining (Bai et al., 2025; Bao et al., 2022; Wu et al., 2024b) or efficient upcycling of dense models in multimodal context (Komatsuzaki et al., 2022; Li et al., 2024; 2025b; Lin et al., 2024; Shu et al., 2024; Wu et al., 2025), aiming to improve efficiency without compromising performance.

**Mixture-of-Experts.** Sparse Mixture-of-Experts (MoE) was introduced to large language models by Gshard (Lepikhin et al., 2020) and refined in subsequent works (Dai et al., 2024; Fedus et al., 2022; Jiang et al., 2024; Wu et al., 2024b; Wei et al., 2024; Xue et al., 2024). This architecture comprises a gating network and a collection of expert subnetworks, typically independent feed-forward modules. Employing a conditional computation strategy, the gating network selectively routes each input token to a sparse subset of these experts for processing. The predominant implementation uses a Top-K routing algorithm (Shazeer et al., 2017; Clark et al., 2022; Fan et al., 2024), wherein the K experts with the highest gating scores are activated for each token. However, despite its efficiency,

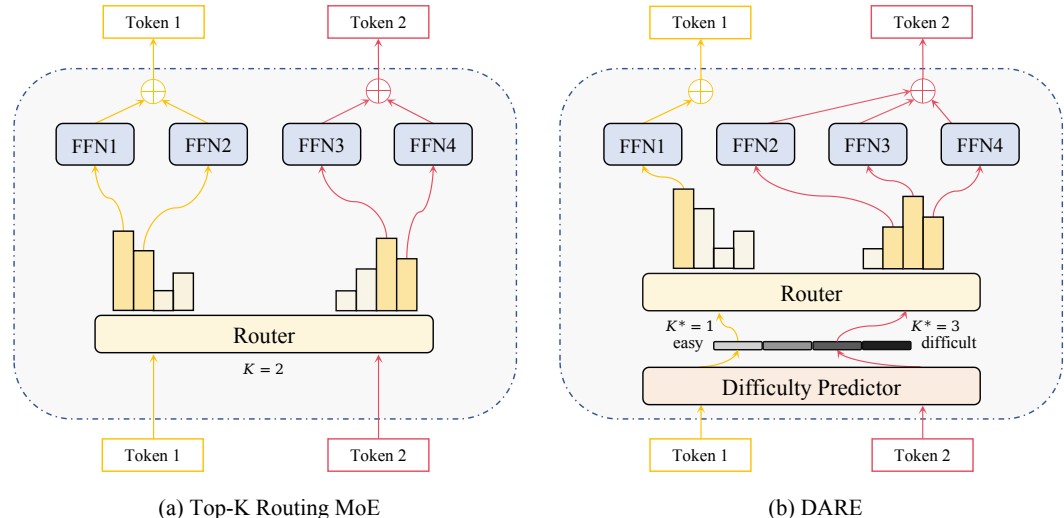

(a) Top-K Routing MoE                                  (b) DARE

Figure 1: Comparison of Top-K routing and DARE: (a) In Top-K routing, each token is routed to a fixed number of top-scoring experts based on a static configuration. (b) In DARE, a learnable difficulty predictor estimates the difficulty of each token, which is then compared against a set of thresholds to adaptively determine the number of experts to activate.

this static allocation assigns a fixed number of experts to all tokens, failing to account for variations in complexity at the token level. This inherent rigidity can constrain the model's expressive power, thus motivating research into more adaptive and dynamic routing strategies.

**Dynamic Expert Allocation.** To overcome the limitations of static expert allocation that ignores token-level complexity, recent studies have proposed various dynamic routing strategies (Huang et al., 2024; Zeng et al., 2024; Jin et al., 2024; Guo et al., 2024; Yue et al., 2024; Wang et al., 2024c). Top-p routing (Huang et al., 2024) adjusts the number of activated experts based on cumulative probability thresholds. AdaMoE (Zeng et al., 2024) and MoE++ (Jin et al., 2024) introduce zero-computation experts to implicitly reduce expert usage for easy tokens. DynMoE (Guo et al., 2024) learns similarity-based thresholds to modulate routing, while Ada-K (Yue et al., 2024) leverages reinforcement learning to predict expert counts. ReMoE (Wang et al., 2024c) replaces Softmax with ReLU gating to activate all positively scored experts. However, these methods still rely on probabilistic sampling, dummy experts, or heuristic rules, lacking an explicit modeling of the relationship between token difficulty and expert allocation. In this work, we take a fundamentally different approach by introducing a learnable difficulty estimator that directly quantifies token-level computation demand, enabling explicit, adaptive, and principle-driven expert routing.

## 3 METHOD

To establish the empirical foundation for our method, we first conduct an analysis demonstrating a strong inverse relationship between token-level perplexity and prediction accuracy. This validates perplexity as an effective proxy for token difficulty. As illustrated in Figure 1, our method introduces two core modules: (1) a lightweight MLP-based difficulty predictor that estimates negative log-likelihoods from hidden representations with an auxiliary MSE loss; and (2) a threshold-based expert allocation mechanism that determines expert counts based on predicted difficulty levels. To accommodate shifting difficulty distributions during training, we apply an online quantile-based threshold adjustment to ensure balanced expert engagement.

### 3.1 PERPLEXITY AS A PROXY FOR DIFFICULTY

A foundational requirement for our adaptive allocation strategy is a reliable and quantifiable metric for token-level difficulty. We propose perplexity (ppl) as a direct proxy for this difficulty, as it intrinsically reflects the model's uncertainty when predicting a token (Brown et al., 2020). Formally, for a

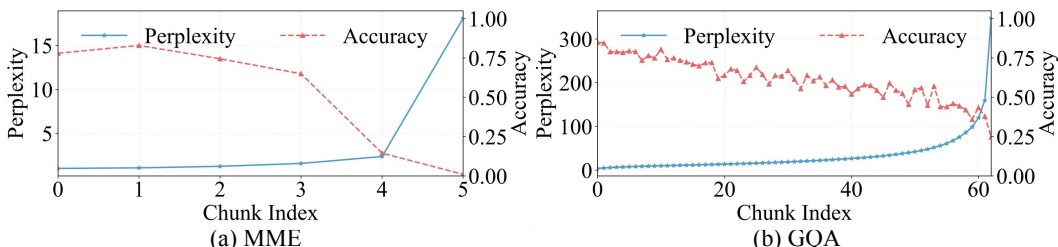

Figure 2: Average prediction accuracy versus token perplexity on MME (Yin et al., 2024) and GQA (Hudson & Manning, 2019), evaluated using MoELLaVA (Qwen2-1.5B). Samples are sorted by ground-truth perplexity and grouped into bins of 200.

token $x_t$ given its preceding context $x_{<t}$ and model parameters $\Theta$, token-level perplexity is defined as the exponentiated negative log-likelihood:

$$\mathrm{ppl}(x_t \mid x_{<t}) = \exp\left(-\log p(x_t \mid x_{<t}; \Theta)\right) \tag{1}$$

A higher perplexity value signifies the model's lower confidence and thus implies greater processing difficulty (Chen et al., 2023; Jiang et al., 2023).

To empirically validate this hypothesis that higher perplexity correlates with greater task difficulty, we perform an analysis on the MME (Yin et al., 2024) and GQA (Hudson & Manning, 2019) datasets. For each sample in the evaluation sets, we calculate the model's prediction correctness and the perplexity of the corresponding ground-truth answer. The samples are then sorted based on their perplexity values and grouped into fixed-size bins of 200. Within each bin, we compute the average accuracy and perplexity to analyze the underlying trends.

As illustrated in Figure 2, the results reveal a strong and consistent negative correlation between average perplexity and prediction accuracy across both datasets. As perplexity increases, the model's accuracy steadily declines. This empirical evidence clearly confirms that perplexity serves as a robust and interpretable proxy for token difficulty, thereby establishing a reliable foundation for our proposed difficulty-aware routing mechanism.

## 3.2 TOKEN DIFFICULTY PREDICTION

Building on our empirical findings, we now formalize the definition of token difficulty. For improved numerical stability and more effective supervision, we employ the logarithmic form of the token-level perplexity from Equation 1, which defines the ground-truth difficulty $d_t$ as the negative log-likelihood:

$$d_t = -\log p(x_t \mid x_{<t}; \Theta) \tag{2}$$

To enable efficient difficulty estimation during inference, we introduce a lightweight difficulty predictor $\mathcal{P}_\phi$, implemented as a multi-layer perceptron (MLP) that takes the token's hidden representation $\mathbf{h}_t$ as input and produces a predicted difficulty value, $\hat{d}_t = \mathcal{P}_\phi(\mathbf{h}_t)$. The predictor is trained jointly with the main model by minimizing the mean squared error (MSE) between the predicted and ground-truth difficulty scores on text tokens over a training batch $\mathcal{B}$:

$$\mathcal{L}_{\mathrm{diff}} = \mathbb{E}_{t \sim \mathcal{B}}\left[(\hat{d}_t - d_t)^2\right] \tag{3}$$

Following standard practices in Mixture-of-Experts training, we incorporate an auxiliary load balancing loss, denoted as $\mathcal{L}_{\mathrm{lb}}$, to promote uniform expert utilization and mitigate expert collapse (Shazeer et al., 2017; Lepikhin et al., 2020). This loss encourages the gating mechanism to distribute tokens more evenly across all experts and is defined as:

$$\mathcal{L}_{\mathrm{lb}} = \frac{N}{|\mathcal{B}|} \sum_{i=1}^{N} f_i \cdot P_i \tag{4}$$

where $N$ denotes the total number of experts, $f_i$ represents the fraction of tokens in the batch routed to expert $i$, and $P_i$ is the average gating probability assigned to expert $i$ across the batch. By introducing a penalty on imbalanced routing patterns, this objective maintains healthy expert diversity and prevents overspecialization (Qiu et al., 2025; Liu et al., 2024a; Shen et al., 2023).

By combining the primary cross-entropy loss ($\mathcal{L}_{\text{CE}}$) (Lin et al., 2024), the proposed difficulty prediction loss ($\mathcal{L}_{\text{diff}}$), and the auxiliary load balancing loss ($\mathcal{L}_{\text{lb}}$), the overall training objective is formally defined as:

$$\mathcal{L} = \mathcal{L}_{\text{CE}} + \alpha \cdot \mathcal{L}_{\text{diff}} + \beta \cdot \mathcal{L}_{\text{lb}} \tag{5}$$

where $\alpha$ and $\beta$ are the weighting hyperparameters. This formulation simultaneously optimizes task performance, difficulty-aware expert routing, and the overall balance of expert utilization.

### 3.3 Online Threshold Adaptation for Expert Allocation

Given the predicted difficulty score $\hat{d}_t$, DARE applies a stratified expert allocation policy to determine the number of experts assigned to each token. For an MoE layer with $M$ experts, we define a set of $M-1$ monotonically increasing thresholds, denoted as $\mathcal{T} = (\tau_1, \tau_2, \ldots, \tau_{M-1})$. The number of experts allocated to token $t$, denoted by $N_e(t)$, is computed as follows:

$$N_e(t) = 1 + \sum_{j=1}^{M-1} \mathbb{I}(\hat{d}_t \geq \tau_j) \tag{6}$$

where $\mathbb{I}(\cdot)$ is the indicator function. This formulation ensures that at least one expert is always selected, while additional experts are progressively activated for tokens with higher predicted difficulty. The token is then routed to the top-$N_e(t)$ experts based on the gating network's scores, allowing the model to dynamically select the most relevant experts for processing.

A key challenge in this setting is that the distribution of predicted difficulty scores $\hat{d}_t$ is non-stationary and evolves throughout training. Relying on fixed thresholds would therefore result in a mismatch between intended and actual expert utilization. To address this issue, we propose an online threshold adaptation mechanism that dynamically adjusts the thresholds to align observed expert usage with a predefined target distribution $\Pi = (\pi_1, \pi_2, \ldots, \pi_M)$, where $\sum_{k=1}^{M} \pi_k = 1$. This target distribution encodes an inductive bias that reflects the empirical long-tailed nature of token difficulty: most tokens are relatively easy and require only limited computation, while a small subset are significantly harder and warrant increased expert capacity (Kandpal et al., 2023). Accordingly, $\Pi$ can be designed to assign higher probability mass to lower expert counts (e.g., $\pi_1 > \pi_2 > \cdots > \pi_M$), thereby promoting efficient and difficulty-aware resource allocation.

At each training step $i$, we begin by computing the empirical cumulative distribution function (CDF), denoted as $F_{\mathcal{B}_i}(\cdot)$, over the predicted token difficulties $\hat{d}_t$ within the current mini-batch $\mathcal{B}_i$. Based on this CDF, we determine the target value for the $j$-th threshold by selecting the quantile corresponding to the cumulative probability $\sum_{k=1}^{j} \pi_k$:

$$\tau_{j,\text{target}}^{(i)} = F_{\mathcal{B}_i}^{-1}\left(\sum_{k=1}^{j} \pi_k\right), \quad \text{for } j = 1, \ldots, M-1 \tag{7}$$

where $F^{-1}$ denotes the inverse CDF (i.e., the quantile function). To ensure more stable optimization and reduce variance from mini-batch fluctuations, the global thresholds are updated via an exponential moving average (EMA):

$$\tau_{j,\text{global}}^{(i)} = \gamma \cdot \tau_{j,\text{global}}^{(i-1)} + (1-\gamma) \cdot \tau_{j,\text{target}}^{(i)} \tag{8}$$

where $\gamma \in [0,1)$ is a momentum coefficient controlling the update rate. This online adaptation mechanism enables the thresholds to continuously track the evolving distribution of token difficulty, facilitating consistent, stable, and complexity-aware expert allocation throughout training.

## 4 Experiments

### 4.1 Experimental Setup

**Model & Training Setup.** Our method builds upon the LLaVA framework (Liu et al., 2024b; 2023c), where 50% of the feed-forward layers are replaced with MoE layers in an alternating layout. We apply DARE only to the MoE layers where visual and textual tokens have already been aligned,

Table 1: Performance comparison of different MoE routing strategies on the Qwen2-1.5B (Team, 2024) backbone across multiple vision-language benchmarks. $N_A$ denotes the average number of activated parameters per inference (in billions), and Avg K indicates the average number of activated experts per MoE layer. Results for MoE++ are marked with $*$ to indicate the inclusion of additional lightweight experts, making both the actual number of activated parameters and Avg K effectively higher than reported. In the table, bold numbers indicate the best performance and underlined numbers denote the second-best.

| Method | $N_A$ | Avg K | MME | MMB | POPE | SQA$^I$ | VQA$^T$ | GQA | VQA$^{v2}$ | MM-Vet |
|---|---|---|---|---|---|---|---|---|---|---|
| *Dense* | | | | | | | | | | |
| LLaVA-1.5 (Vicuna-13B) | 13B | - | 1531.3 | 67.7 | 85.9 | 71.6 | 61.3 | 63.3 | 80.0 | 35.4 |
| LLaVA-1.5 (Vicuna-7B) | 7B | - | 1510.7 | 64.3 | 85.9 | 66.8 | 58.2 | 62.0 | 78.5 | 30.5 |
| LLaVA-Phi (Phi-2-2.7B) | 2.7B | - | 1335.1 | 59.8 | 85.0 | 68.4 | 48.6 | - | 71.4 | 28.9 |
| *Sparse* | | | | | | | | | | |
| MoELLaVA (Lin et al., 2024) | 2.13B | 2 | 1369.2 | 65.2 | 86.2 | **69.9** | 56.8 | 61.5 | **79.6** | 26.7 |
| MoE++ (Jin et al., 2024) | 1.49B$^*$ | 0.90$^*$ | 1348.3 | 64.3 | 85.5 | 68.4 | 55.7 | 61.3 | 79.4 | 26.7 |
| DYNMoE (Guo et al., 2024) | 1.55B | 1.00 | 1309.3 | 63.1 | 81.9 | 67.0 | 49.9 | 57.7 | 76.4 | 25.6 |
| Top-p (Huang et al., 2024) | 1.95B | 1.69 | 1378.2 | 65.6 | 85.7 | 69.5 | 56.2 | 61.6 | 79.1 | **28.8** |
| ReMoE (Wang et al., 2024c) | 1.89B | 1.59 | 1347.1 | 65.5 | **86.5** | 69.8 | 55.2 | **62.5** | 79.6 | 28.5 |
| **DARE(ours)** | 1.68B | 1.22 | **1404.3** | 66.1 | 86.0 | 69.9 | 57.0 | 62.0 | 79.6 | 28.8 |

to enable dynamic, difficulty-aware expert selection. We adopt the three-stage training procedure from MoELLaVA (Lin et al., 2024), using LLaVA-1.5-558k (Liu et al., 2024b), SViT (Zhao et al., 2023), LVIS (Wang et al., 2023), LRV (Liu et al., 2023a), and MIMIC-IT (Li et al., 2025a) for Stage 1& 2 training. In Stage 3, we fine-tune MoE-LLaVA with DARE on LLaVA-mix-665k (Liu et al., 2024b). Further configurations are detailed in the Appendix D.

**Evaluation & Baselines.** We evaluate our approach on a comprehensive benchmark suite, including MME (Yin et al., 2024), MMB (Liu et al., 2024c), VQA-v2 (Goyal et al., 2017), GQA (Hudson & Manning, 2019), TextVQA (Singh et al., 2019), MM-Vet (Yu et al., 2023), ScienceQA (Lu et al., 2022), and POPE (Li et al., 2023c). To ensure fair comparison, all experiments use the same backbone models and training data. Results for the dense baseline and MoE-LLaVA (Lin et al., 2024) on StableLM are from (Lin et al., 2024), and DYNMoE (Guo et al., 2024) is reproduced using the official code. Other methods (Top-p (Huang et al., 2024), MoE++(Jin et al., 2024), ReMoE(Wang et al., 2024c)) on StableLM, and all methods on Qwen2 (Team, 2024)/Qwen3 (Yang et al., 2025), are reimplemented under a unified framework. Appendix D includes full implementation details and evaluation metrics, providing all the necessary information to reproduce our experiments and verify the reported results.

## 4.2 MAIN RESULTS

**Comparison with State-of-the-Art MoE Routing Strategies.** Table 3 presents a comparative analysis of our DARE method against several established MoE routing baselines on the Qwen2-1.5B backbone. The results reveal a consistent trade-off between model performance and computational cost across existing strategies. Static routing methods such as Top-2 in MoELLaVA and dynamic approaches like Top-p and ReMoE achieve strong performance on most benchmarks, but incur substantial computational overhead by activating between 1.6 and 2 experts per token on average. For example, ReMoE and Top-p require 1.89B and 1.95B activated parameters, respectively. In contrast, efficiency-oriented methods such as MoE++ and DYNMoE reduce the average number of activated experts to approximately 1.0 or fewer, but often suffer from degraded performance on key benchmarks including MME and MMBench. DYNMoE, in particular, shows a notable drop in accuracy, likely due to a simplistic dynamic allocation mechanism that does not fully leverage the model's expert capacity. In comparison, DARE achieves a more favorable balance between accuracy and efficiency. It consistently matches or exceeds the performance of high-cost methods like Top-p and ReMoE, achieving the best results on MME (1404.3), MMBench (66.1), and MM-Vet (28.8), while activating only 1.22 experts and 1.68B parameters on average. This corresponds to a reduction of over 21 percent in activated parameters compared to the standard Top-2 baseline. Notably, our sparse model rivals much larger dense architectures, outperforming the 2.7B LLaVA-Phi across almost all benchmarks with only a fraction of the parameters. These results highlight the effectiveness

of DARE's difficulty-aware routing strategy in enabling informed expert selection that enhances computational efficiency without sacrificing performance.

**Results on Different LLM Backbones.** To evaluate the generalization capability and robustness of our proposed routing strategy, we extend our experiments to multiple LLM backbones by integrating our method into both StableLM-1.6B and Qwen3-1.7B. As shown in Table 2, we compare the performance of our approach with the standard Top-K routing baseline for each model. The results demonstrate that the effectiveness of our method is not limited to a specific architecture. Across both model families, our difficulty-aware routing strategy consistently outperforms the

Table 2: Performance comparison on different backbone models. S-1.6 refers to StableLM-1.6B and Q3-1.7 refers to Qwen3-1.7B. Top-K serves as the baseline routing strategy. The average number of activated experts is 1.94 for S-1.6B and 1.26 for Q3-1.7B.

| Method | LLM | VQA$^T$ | MMB | MME | MM-Vet |
|--------|-----|---------|-----|-----|--------|
| Top-K | S-1.6 | 50.1 | 60.2 | 1318.2 | 26.9 |
| **Ours** | | **50.5** | **61.0** | **1345.9** | **27.8** |
| Top-K | Q3-1.7 | 59.9 | 67.6 | 1406.6 | 30.3 |
| **Ours** | | **60.0** | **68.8** | **1437.1** | **31.9** |

baseline, yielding notable improvements on all evaluated benchmarks. Importantly, these gains are achieved with high computational efficiency. The average number of activated experts (1.94 for StableLM-1.6B and 1.26 for Qwen3-1.7B) remains lower than that of the conventional Top-2 routing, indicating that the performance improvement stems from more informed expert selection rather than increased computation. This consistent superiority across diverse model backbones highlights the strong generalization capability and adaptability of our method, establishing it as a broadly applicable and computationally efficient routing enhancement for MoE-based language models.

Table 3: Performance comparison on the Qwen2-0.5B backbone with an expanded expert configuration (16 experts in total). This setting evaluates the scalability of different MoE routing strategies. DARE maintains superior performance while activating fewer experts on average, demonstrating efficient utilization of increased expert capacity.

| Method | Avg K | MME | MMB | POPE | SQA$^I$ | VQA$^T$ | GQA | VQA$^{v2}$ | MM-Vet |
|--------|-------|-----|-----|------|---------|---------|-----|------------|--------|
| MoELLaVA (Lin et al., 2024) | 2 | 1198.7 | 53.9 | **86.8** | 59.4 | 45.8 | 57.8 | 75.1 | 25.0 |
| **DARE(ours)** | 1.56 | **1267.6** | **56.4** | 86.3 | **61.5** | **47.8** | **59.3** | **76.0** | **26.4** |

**Generalization to Expanded Expert Capacity.** To further assess the scalability of our DARE routing strategy, we conduct experiments with an expanded expert configuration, using a smaller Qwen2-0.5B backbone but increasing the total number of experts to sixteen. This setup examines the model's ability to effectively exploit a larger and more granular expert pool under limited parameter budgets, a key challenge for scalable MoE systems. As shown in Table 3, DARE demonstrates strong adaptability in this more demanding setting. Compared to the baseline MoELLaVA, which activates 2 experts per layer, DARE achieves higher accuracy across nearly all benchmarks while reducing the average number of activated experts to 1.56. It yields notable improvements on multimodal reasoning tasks such as MME (**+68.9**) and MMBench (**+2.5**), along with consistent gains on SQA$^I$, VQA$^T$, and GQA, all without increasing computational cost. These results highlight DARE's ability to efficiently manage expanded expert diversity and leverage finer-grained specialization through adaptive difficulty-aware routing. Overall, DARE scales favorably with larger expert pools, maintaining both accuracy and efficiency, and confirming its robustness as a general routing principle that remains effective as model capacity and expert granularity continue to increase.

## 4.3 ABLATION STUDY

**Impact of the Difficulty Proxy.** We perform an ablation study to examine the effect of different difficulty proxies on routing behavior. We evaluate two baseline difficulty proxies, including a random signal and a gating-entropy–based measure that captures the routing network's internal uncertainty, and compare them against our log-perplexity–based proxy. As shown in Table 4, both alternatives underperform our log-perplexity–based proxy across all benchmarks. The gating entropy signal yields marginal or even degraded results compared to random routing, suggesting that internal gating uncertainty alone is insufficient to capture token-level complexity. In contrast, our log-perplexity proxy provides a semantically grounded measure of difficulty, enabling more informed expert activation and leading to consistent gains on VQA$^T$, MMBench, MME, and SQA.

These findings highlight that the benefits of DARE stem not merely from dynamic expert allocation, but from a principled estimation of token difficulty that guides routing decisions toward more efficient and effective computation.

Table 4: Ablation on the difficulty proxy. We compare our perplexity-based proxy against random signal and gating entropy based signal, showing its effect on guiding expert allocation.

| Difficulty Proxy | VQA$^T$ | MMB | MME | SQA$^I$ |
|---|---|---|---|---|
| Random | 56.3 | 65.3 | 1380.2 | 69.2 |
| Gating Entropy | 55.5 | 65.1 | 1352.6 | 69.2 |
| **Log ppl (Ours)** | **57.0** | **66.1** | **1404.1** | **69.9** |

Table 5: Ablation on the threshold adaptation mechanism. We compare online adaptation against batch thresholds, highlighting the benefit of dynamic adjustment.

| Thres Strategy | VQA$^T$ | MMB | MME | MM-Vet |
|---|---|---|---|---|
| Batch | 55.5 | 65.6 | 1381.0 | 27.1 |
| **Online(Ours)** | **57.0** | **66.1** | **1404.1** | **28.8** |

**Impact of Online Threshold Adaptation.** To evaluate the effectiveness of our threshold adaptation strategy, we compare the proposed online thresholding mechanism with a batch-level threshold baseline. As shown in Table 5, the online strategy consistently outperforms the batch alternative across all benchmarks, with notable gains in MME (+23.1) and MM-Vet (+1.7). These results indicate that dynamically adjusting thresholds over time not only leads to more stable and adaptive expert allocation but also better aligns computation with input complexity, thereby improving both efficiency and overall performance in multimodal tasks.

**Impact of Predefined Target Distribution.**
Table 6 examines the effect of the predefined target distribution $\Pi$, which governs the proportion of tokens routed to different numbers of activated experts. We evaluate three variants: a uniform distribution, an inverse distribution that favors activating more experts, and our proposed long-tailed distribution. The uniform and inverse strategies yield inconsistent or degraded performance, indicating that naively flattening the routing probabilities or biasing toward heavier computation fails to align expert usage with token difficulty. In contrast, our long-tailed distribution achieves the best results

Table 6: Ablation on the predefined target distribution $\Pi$. We compare uniform, inverse, and long-tailed variants. The long-tailed distribution yields consistently superior performance, highlighting its effectiveness in guiding computation-aware expert activation.

| Distribution $\Pi$ | VQA$^T$ | MMB | MME | SQA$^I$ |
|---|---|---|---|---|
| Uniform | 56.5 | 65.0 | 1352.5 | 68.2 |
| Inverse | 55.3 | 65.8 | 1378.2 | 69.1 |
| **Long-tailed(Ours)** | **57.0** | **66.1** | **1404.1** | **69.9** |

across all benchmarks, suggesting that allocating most tokens to a small, efficient subset of experts while reserving larger ensembles for the most challenging inputs provides a more computation-aware and semantically aligned routing prior. This highlights the importance of designing principled target distributions when shaping expert activation patterns in sparse MoE architectures.

## 4.4 VISUALIZATION AND ANALYSIS

**Analysis of Expert Allocation Patterns.** We visualize the average number of activated experts per MoE layer on DARE(Qwen3-1.7B) across eight benchmarks in Figure 3 to examine how DARE allocates computational resources in a task- and layer-adaptive manner. The results reveal that expert activation varies significantly across both tasks and layers. More complex reasoning tasks such as ScienceQA and TextVQA elicit higher average top-k values, while perception-focused tasks like MME and POPE require fewer active experts. Moreover, expert usage trends higher in the deeper layers, which aligns with the hierarchical processing in Transformers where these later stages are responsible for more complex semantic and abstract reasoning (Geva et al., 2020), thus benefiting from greater specialized capacity. This dynamic allocation reflects DARE's ability to recognize contextual difficulty and modulate expert selection accordingly. Notably, the overall number of active experts remains well below the fixed Top-2 baseline, demonstrating that DARE achieves efficient computation without sacrificing performance. These findings highlight the model's capacity to balance flexibility and efficiency through fine-grained, difficulty-aware routing.

**Analysis of Expert Load Balance.** To evaluate the routing behavior of different dynamic MoE strategies, we visualize the expert activation frequencies for ReMoE, Top-P, and our proposed DARE in Figure 4. All three methods incorporate a standard load-balancing loss (Lepikhin et al., 2020) to

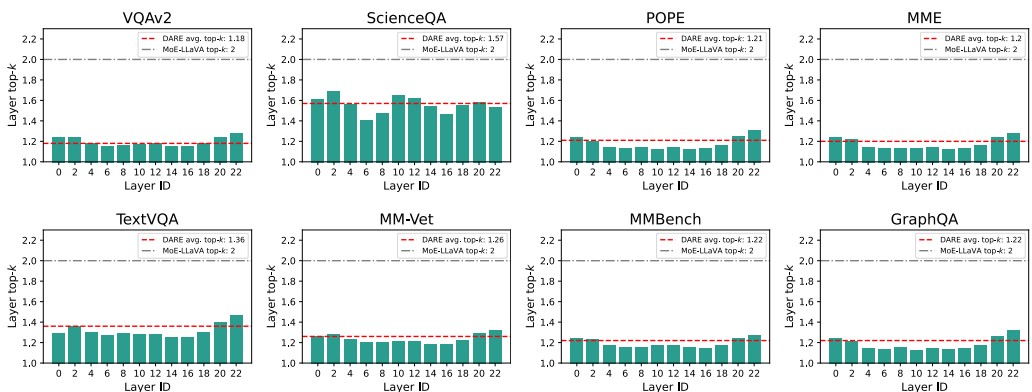

Figure 3: Average top-k activated experts of DARE on vision-language benchmarks. We report the average top-k expert activations per MoE layer using Qwen3-1.7B as the language model backbone, highlighting how the routing mechanism adapts to different input complexities across layers.

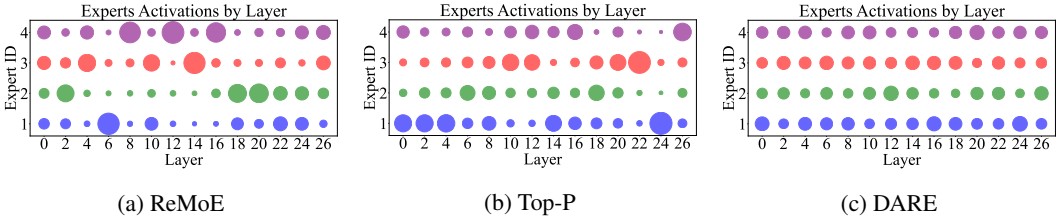

Figure 4: Expert activations by layer for three MoE configurations: (a) ReMoE, (b) Top-P, and (c) DARE. The x-axis represents the layer number, and the y-axis corresponds to the expert ID. The size of each circle reflects the activation strength of the corresponding expert at each layer.

encourage uniform expert utilization. The visualizations reveal a significant expert load imbalance in both ReMoE and Top-P, where a chronic over-reliance on a few hot experts indicates a tendency toward expert collapse (Wang et al., 2024a). This skewed utilization undermines the intended capacity expansion of MoE architectures and can degrade training stability and model generalization. In sharp contrast, DARE demonstrates a remarkably uniform distribution of expert activations, suggesting it successfully mitigates expert collapse. By replacing a static selection policy with a dynamic, difficulty aware allocation strategy, DARE prevents the model from defaulting to a small set of dominant experts, thereby promoting a more balanced and diverse utilization of the entire expert pool. This balanced routing is crucial for maintaining the scalability and efficiency of MoE models, ultimately enabling a more effective use of model capacity and leading to improved performance.

**Analysis of Modality-Generalist Expert Behavior.** To examine how experts process inputs from different modalities, we analyze the routing distributions for text and image tokens separately on the Qwen2-1.5B backbone, as shown in Figure 5. The results indicate that the expert activation patterns for text and image tokens are highly similar across all layers. This suggests that the model does not induce strong modality-specific specialization among experts. Instead, each expert exhibits the capacity to handle both textual and visual information, functioning as a modality-generalist. Such behavior is beneficial for multimodal tasks, as it supports

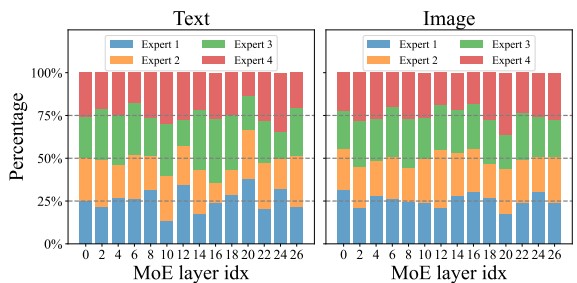

Figure 5: Distribution of modalities across different experts. Interrupted lines mean a perfectly balanced distribution of tokens.

more flexible and unified information processing. By promoting a shared expert pool rather than

segregated modality-specific pathways, DARE facilitates deeper cross-modal interaction and enables more effective reasoning over jointly represented visual and textual inputs.

**Analysis of Inference efficiency.** We evaluate the inference efficiency of the proposed DARE against the MoE-LLaVA baseline, which adopts a standard top-2 gating mechanism. The detailed results are provided in Table 7. In terms of memory consumption, DARE requires 7.09 GB, essentially matching MoE-LLaVA's 7.10 GB, demonstrating that the introduced difficulty predictor adds negligible memory overhead. More importantly, DARE delivers clear computational advantages: it reduces inference FLOPs per token by 41.5 % (from 89.28 GFLOPs to 52.25 GFLOPs), which directly translates into a 27 % increase in throughput (75 vs. 59 tokens/s) and a 13 % reduction in wall-clock time per sample (5.4 s vs. 6.2 s). These results confirm that DARE substantially accelerates MoE inference while maintaining the same memory footprint, underscoring its effectiveness and practical utility. Additional detailed results of efficiency comparisons across various backbones are provided in Table 9.

Table 7: Efficiency comparison of DARE versus MoE-LLaVA on Qwen2-1.5B. MoE-LLaVA results are obtained with DeepSpeed's top-2 gating. Symbols ↓ and ↑ indicate that lower or higher values are better, respectively. Reported numbers are the mean of five independent runs.

| Model | Memory ↓ (GB) | Inference FLOPs ↓ (GFLOPs/token) | Throughput ↑ (token / second) | Wall-clock Time ↓ (second / sample) |
|---|---|---|---|---|
| MoE-LLaVA | 7.10 | 89.28 | 59 | 6.2 |
| DARE(Ours) | 7.09 | 52.25 | 75 | 5.4 |

## 5 CONCLUSION

In this work, we present DARE, a novel routing strategy for Mixture-of-Experts models that explicitly incorporates token-level difficulty into expert allocation. By introducing a lightweight difficulty predictor and a threshold-based mechanism for dynamic expert selection, DARE enables fine-grained routing that better aligns computational resources with the intrinsic complexity of input tokens. Empirical results on vision-language tasks validate our approach, showing that DARE outperforms both static and existing dynamic routing baselines in performance and efficiency. Our findings underscore the importance of token difficulty modeling in expert selection, paving the way for more interpretable, resource-efficient, and scalable MoE architectures. Future work includes exploring alternative difficulty estimations and extend DARE to large-scale pre-training to further evaluate its scalability and effectiveness in language modeling tasks.

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

# APPENDIX

## A ETHICS STATEMENT

Our study does not involve human or animal subjects, personal or sensitive data, or any procedures that could raise privacy, security, or legal concerns. All datasets used are publicly available and widely adopted in the research community. We have carefully checked to avoid introducing discrimination, bias, or other harmful effects in our methodology and applications.

## B USE OF LARGE LANGUAGE MODELS

Large language models (e.g., ChatGPT) were used exclusively to polish the manuscript, including grammar correction, clarity improvements, and overall readability. Their role was limited to language editing; they did not participate in the conception of the research, experimental design, data analysis, or interpretation of results. All suggested edits were reviewed and incorporated at the discretion of the authors.

- **Grammar and mechanics**: correcting grammar, punctuation, and typographical errors.
- **Clarity and concision**: rewriting sentences to improve clarity, enhancing paragraph flow, and reducing redundancy.
- **Captions and abstracts**: refining figure and table captions and polishing the abstract for clarity and readability.
- **Formatting suggestions**: minor LaTeX and presentation adjustments to improve overall readability and layout.

## C REPRODUCIBILITY

We provide all implementation details necessary to reproduce our results. The main paper describes the model architecture, training objectives, and evaluation protocols in detail, while the Appendix D presents complete hyperparameter settings and other experimental configurations to facilitate faithful replication.

## D EXPERIMENTS SETTING

### D.1 IMPLEMENTATION DETAILS

Our method is built upon the LLaVA framework (Liu et al., 2024b; 2023c), wherein 50% of the feed-forward layers are replaced by Mixture-of-Experts (MoE) layers in an alternating configuration. We apply DARE exclusively to the MoE layers where visual and textual tokens have already been aligned, thereby enabling dynamic, difficulty-aware expert selection. For different backbone models, the specific layers where DARE is applied are as follows: layers 2 to 26 for Qwen2 (Team, 2024); layers 10 to 22 for StableLM; and all MoE layers for Qwen3 (Yang et al., 2025).

The hyperparameters for DARE are set as follows: $\alpha = 1$ and $\beta = 0.01$. For the token difficulty distribution $\Pi$, we adopt model-specific priors: for Qwen2-1.5B, $\pi_1 = 0.6$, $\pi_2 = 0.3$, $\pi_3 = 0.09$, and $\pi_4 = 0.01$; for StableLM-1.6B, $\pi_1 = 0.4$, $\pi_2 = 0.3$, $\pi_3 = 0.2$, and $\pi_4 = 0.1$; and for Qwen3-1.7B, $\pi_1 = 0.5$, $\pi_2 = 0.3$, $\pi_3 = 0.15$, and $\pi_4 = 0.05$. We set $\tau_{1,\text{global}}^{(0)} = 0, \tau_{2,\text{global}}^{(0)} = 1, \tau_{3,\text{global}}^{(0)} = 2$.

For visual encoding, we adopt CLIP-336 (Radford et al., 2021) to maintain consistency with the strongest baseline MoE-LLaVA (Lin et al., 2024) in the 2B parameter regime. For Qwen2-1.5B and Qwen3-1.7B, we employ SigLIP-so-384 (Zhai et al., 2023) as the vision encoder to further exploit the benefits of our approach. The visual encoder is connected to the language model via a lightweight projector consisting of two linear layers separated by a GELU activation (Hendrycks & Gimpel, 2016). All other training configurations are shown in Table 8.

**Difficulty Predictor.** The difficulty predictor is a lightweight feed-forward network that maps each token's hidden representation to a non-negative difficulty score. Given hidden states $h \in \mathbb{R}^{B \times L \times H}$, the input is first normalized by an RMSNorm layer. The normalized vector is then projected from dimension $H$ to a hidden dimension $d$ (we choose $d = 256$) through a fully connected layer, followed by a SiLU activation and a dropout layer with rate $0.1$ to mitigate overfitting. A second linear layer reduces the hidden dimension to $1$, and a final Softplus activation guarantees a positive output.

$$\hat{d} = \text{Softplus}\big(W_2 \, \text{Dropout}\big[\text{SiLU}(W_1 \, \text{RMSNorm}(h))\big]\big), \tag{9}$$

Here, $W_1 \in \mathbb{R}^{d \times H}$ and $W_2 \in \mathbb{R}^{1 \times d}$. The resulting $\hat{d} \in \mathbb{R}^{B \times L}$ provides a non-negative difficulty score for each token.

Table 8: Detailed training hyper-parameters and configuration.

| Config | Models | | |
|---|---|---|---|
| | StableLM | Qwen2 | Qwen3 |
| Expert Numbers | | 4 | |
| Deepspeed | | Zero2 | |
| Data | | LLaVA-Finetuning | |
| Image resolution | $336 \times 336$ | $384 \times 384$ | $384 \times 384$ |
| Image encoder | CLIP-Large/336 | SigLIP-so-384 | SigLIP-so-384 |
| Image projector | | Linear layers with GeLU | |
| Epoch | | 1 | |
| Learning rate of backbone | | 2e-5 | |
| Learning rate of difficulty predictor | | 1e-4 | |
| Learning rate schedule | | Cosine | |
| Weight decay | | 0.0 | |
| Batch size per GPU | 2 | 2 | 1 |
| Precision | | bf16 | |

## D.2 TRAINING PROTOCOL

Our training strategy follows a three-stage pipeline inspired by MoE-LLaVA, with each stage tailored to progressively enhance multi-modal capabilities:

### D.2.1 STAGE I: VISION-LANGUAGE ALIGNMENT.

This stage aims to adapt visual features into the language model's embedding space to establish multi-modal understanding. We utilize the LLaVA 1.5-558k (Liu et al., 2023b) dataset to train a projection layer that maps CLIP and SigLIP features into the token space of the LLM. Here, image patches are treated as pseudo-text tokens. During this stage, all parameters are frozen except for the projection layer, and the model retains a fully dense architecture.

### D.2.2 STAGE II: DENSE MODEL BOOTSTRAPPING.

We further train the dense model using a curated hybrid dataset composed of SViT (Zhao et al., 2023), LVIS (Wang et al., 2023), LRV (Liu et al., 2023a), and MIMIC-IT (LA split only) (Li et al., 2023a). This phase emphasizes learning from complex, instruction-based tasks involving image reasoning and textual comprehension. The vision encoder remains frozen to preserve the integrity of visual representations, while the language model and projection layers are fine-tuned.

### D.2.3 STAGE III: SPARSE EXPERT TRAINING.

In the final stage, we convert the dense model into the proposed DARE architecture by replacing selected FFNs with MoE layers. Each expert is initialized by duplicating the corresponding dense FFN weights. The entire model is then fine-tuned using the LLaVA-mix-665k dataset (Liu et al.,

2023b), following the same procedure as MoE-LLaVA, to effectively train the sparse expert routing mechanism.

### D.3 Evaluation & Baseline

#### D.3.1 Evaluation.

Our evaluation framework spans multiple benchmarks to assess comprehensive multimodal capabilities. We employ MME (Yin et al., 2024) and MMB (Liu et al., 2024c), which contain diverse sub-tasks for measuring visual understanding and reasoning proficiency. To thoroughly evaluate question-answering performance across different domains, we utilize several specialized VQA datasets: VQA-v2 (Goyal et al., 2017) and GQA (Hudson & Manning, 2019) for testing everyday visual comprehension and relationship inference; TextVQA (Singh et al., 2019) for assessing the model's ability to interpret textual elements embedded within images; and ScienceQA (Lu et al., 2022) for evaluating scientific knowledge integration. (Li et al., 2023c) for evaluating object hallucination.

Additionally, to quantify the evenness of token-expert allocation in MoE models, we introduce the average k values (Avg K) metric which measures the dispersion of expert utilization relative to mean workload. For a batch of input sequences, we compute the average count of experts that each token activates. we track how many experts are allocate to each token, resulting in a allocation vector $\mathbf{t} = [t_1, t_2, ..., t_N]$, where $t_i$ represents the number of experts allocated to token $i$ and $N$ is the total number of tokens. The average k values is then calculated as:

$$\text{Avg K} = \text{mean}(\mathbf{t}) \tag{10}$$

#### D.3.2 Baselines.

In this section, we provide the reproduction details for our selected baselines. All reproduced method use Qwen2 1.5B as backbone and SigLIP-so-384 as vision encoder. All reproduced method follow the training pipeline detailed in Section: Training Protocol.

**Top-p (Huang et al., 2024):** We reproduce the Top-p routing strategy based on the original description. Although it was originally proposed for language models with 16 experts, our setup uses only 4 experts. Directly applying the original settings in this smaller-expert regime leads to a collapse in expert selection, where no experts are activated. To address this, we set the dynamic loss coefficient to $3 \times 10^{-7}$ and adjust the top-p threshold to 0.9999, ensuring that a sufficient number of experts are selected throughout training.

**DynMoE (Guo et al., 2024):** We reproduce DynMoE based on the official codebase provided by the authors. All configurations and training protocols remain unchanged. The only modification we make is implementing the necessary code to enable compatibility with the Qwen2 model.

**MoE++ (Jin et al., 2024):** We follow the original implementation of MoE++ without any modifications. The model uses one Zero Expert, one Copy Expert, and two Constant Experts, consistent with the configuration described in the original paper.

**ReMoE (Wang et al., 2024c):** We reproduce ReMoE based on the original configuration and official codebase. However, directly applying the original settings in our experimental setup leads to a collapse in expert selection, where no experts are activated. To address this issue, we follow the authors' implementation and set the `top_k` parameter to 500, which stabilizes the expert routing during training.

## E Limitations and Future Work

### E.1 Limitations

**Dependence on a Predefined Target Distribution $\Pi$.** A primary limitation of DARE lies in the reliance of its online threshold adaptation mechanism on a predefined target expert distribution, $\Pi$. In our current implementation, $\Pi$ is treated as a discrete, model-specific hyperparameter that must be manually tuned. Although this setting works well in our experiments, it introduces additional

effort and potential sensitivity to the chosen value. The lack of a continuous, learnable function for $\Pi$ may require extra tuning when adapting DARE to new architectures or tasks.

**Scalability to Large-Scale Pre-training.** Another limitation concerns scalability. Due to computational constraints, we evaluated DARE primarily in fine-tuning scenarios with models up to 1.7B parameters. We have not yet explored its behavior when integrated into large-scale pre-training from scratch. Consequently, its training stability, convergence dynamics, and ultimate impact on model capabilities in such pre-training settings remain unverified.

### E.2 FUTURE WORK

To address these issues, we plan three complementary research directions. First, to reduce the need for manual tuning of $\Pi$, we will explore methods to automatically learn or adapt this distribution. Possible approaches include modeling $\Pi$ with continuous parametric forms to capture the long-tailed nature of token difficulty, or designing an adaptive mechanism that adjusts $\Pi$ based on training signals such as loss trends or gradient statistics. Such techniques would enhance DARE's autonomy and generalization across tasks and architectures.

Second, we will extend DARE to large-scale pre-training. As resources permit, we aim to integrate DARE into the pre-training of substantially larger MoE models to systematically evaluate its scalability, stability, and efficiency in realistic deployment scenarios.

Finally, we plan to explore alternative difficulty proxies. While log-perplexity proved effective in this study, a single metric may not fully capture token-level difficulty. Future work will investigate hybrid proxies that incorporate measures of model uncertainty or token-level gradient norms, enabling a more comprehensive and robust difficulty estimation.

## F ADDITIONAL EXPERIMENTAL AND VISUALIZATION RESULTS

### F.1 EFFICIENCY EVALUATION

As shown in Table 9, we evaluate the efficiency of DARE on three different backbones. The results indicate that DARE maintains a memory footprint comparable to the MoE-LLaVA baseline, introducing negligible computational overhead. In contrast, it consistently delivers higher throughput and lower latency across all backbones by markedly reducing FLOPs and MACs, underscoring its superior inference efficiency.

Table 9: Efficiency comparison of DARE versus MoE-LLaVA on three backbones. MoE-LLaVA results are obtained with DeepSpeed's top-2 gating. Symbols ↓ and ↑ indicate that lower or higher values are better, respectively. Reported numbers are the mean of five independent runs.

| Backbone | Model | Memory ↓ (GB) | Inference FLOPs ↓ (GFLOPs/token) | Inference MACs ↓ (GMACs/token) | Throughput ↑ (token / second) | Wall-clock Time ↓ (second / sample) |
|---|---|---|---|---|---|---|
| Qwen2-1.5B | MoE-LLaVA | 7.10 | 89.28 | 44.64 | 59 | 6.2 |
| | DARE(Ours) | 7.09 | 52.25 | 26.12 | 75 | 5.4 |
| Stablelm-1.6B | MoE-LLaVA | 6.46 | 53.20 | 26.59 | 82 | 2.8 |
| | DARE(Ours) | 6.48 | 42.96 | 21.48 | 97 | 2.5 |
| Qwen3-1.7B | MoE-LLaVA | 7.16 | 71.36 | 35.68 | 70 | 6.8 |
| | DARE(Ours) | 7.15 | 52.26 | 26.13 | 88 | 6.2 |

### F.2 ROUTING DISTRIBUTIONS

In this section, we present the routing distributions of our proposed method DARE, applied to three backbone models: Qwen2-1.5B, Qwen3-1.7B, and StableLM-1.6B. The evaluation is conducted across six benchmarks: MME (Yin et al., 2024), GQA (Hudson & Manning, 2019), TextVQA (Singh et al., 2019), MM-Vet (Yu et al., 2023), ScienceQA (Lu et al., 2022), and POPE (Li et al., 2023c). As illustrated in Figure 6,7,9 and 10, both Qwen2 and Qwen3 exhibit remarkably consistent and well-balanced expert utilization across layers, indicating that DARE effectively enforces strong load balancing. Moreover, this equilibrium is preserved across both image and text modalities, suggesting

that DARE induces modality-agnostic routing behavior without relying on explicit modality-aware components.

As shown in Figure 8 and 11, the routing distribution on StableLM-1.6B also demonstrates overall balance in expert load. However, a closer inspection reveals modality-dependent specialization: Experts 1 and 2 are predominantly selected for image inputs, while Experts 3 and 4 show a clear preference for textual inputs. Interestingly, this emergent specialization occurs despite using a shared and unified routing mechanism, and may reflect inductive biases intrinsic to the StableLM architecture or its pretraining data. Importantly, this form of specialization does not compromise global load balance or lead to expert collapse, but instead represents a structured divergence that remains within the bounds of efficient expert allocation. These findings highlight DARE's robustness across diverse model architectures, as well as its capacity to adaptively elicit either generalized or modality-sensitive expert behaviors depending on the underlying backbone.

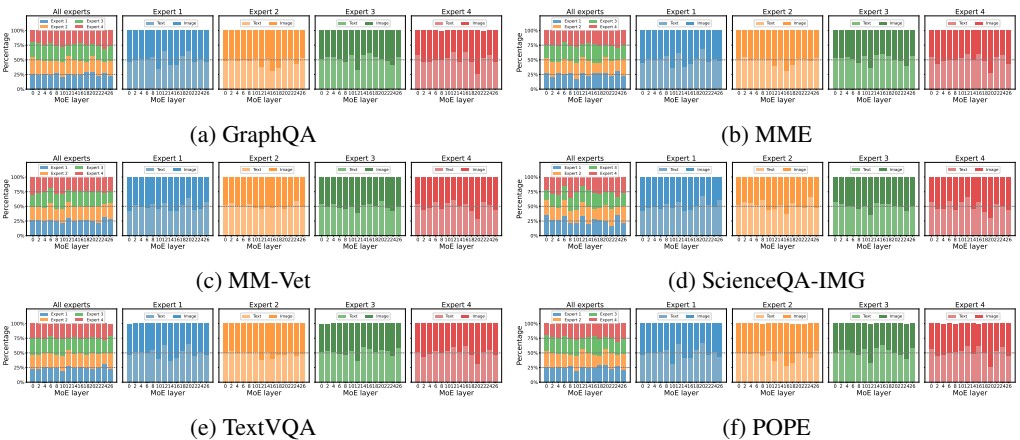

Figure 6: Distribution of expert loadings and preferences on DARE (Qwen2-1.5B). Expert utilization remains well balanced across modalities and layers, indicating stable and uniform routing.

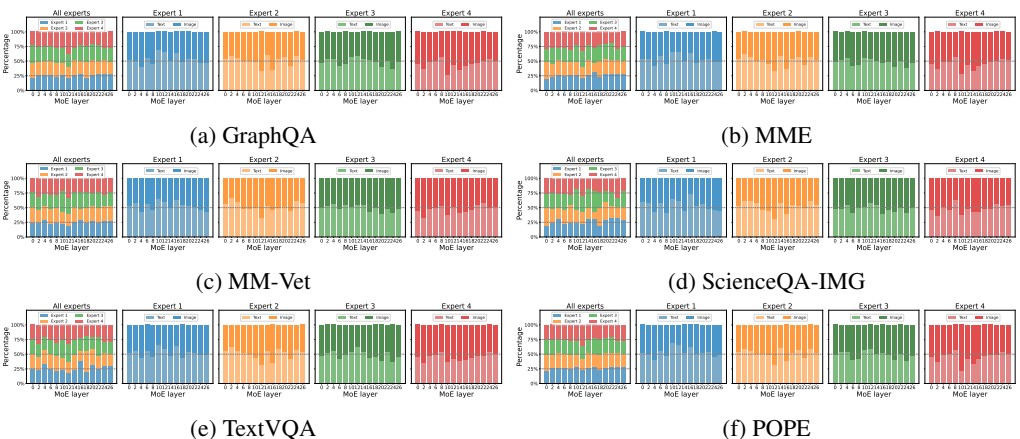

Figure 7: Distribution of expert loadings and preferences on DARE (Qwen3-1.7B). Expert utilization remains well balanced across modalities and layers, indicating stable and uniform routing.

### F.3 EXPERT SIMILARITY MATRIX ACROSS LAYERS

As shown in Figure 12,13 and 14, to examine the diversity of experts, we compute a layer-wise expert similarity matrix for DARE across multiple model architectures, including Qwen2-1.5B, Qwen3-1.7B, and StableLM-1.6B. For each layer, we record the cosine similarity between every pair of experts at test time. Across all three models, the inter-expert cosine similarities remain consistently

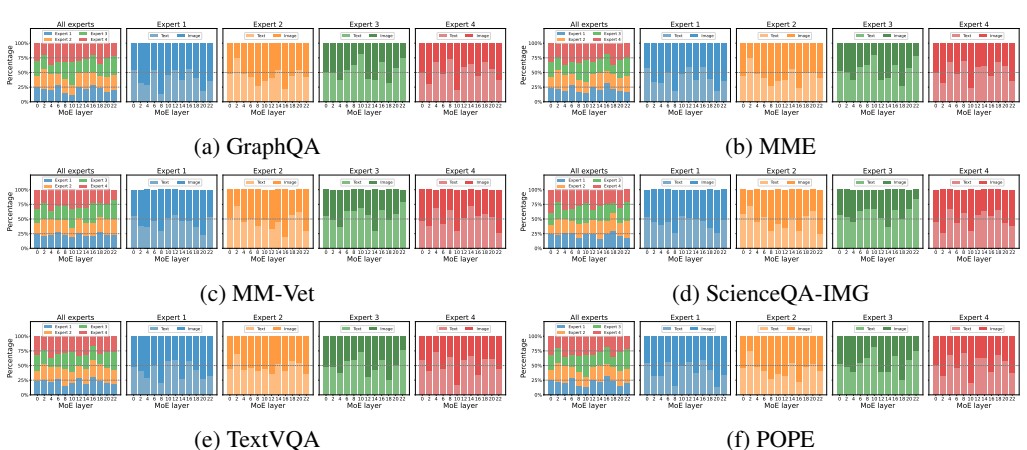

Figure 8: Distribution of expert loadings and preferences on DARE(Stablelm-1.6B). Expert utilization remains well balanced across modalities and layers, indicating stable and uniform routing.

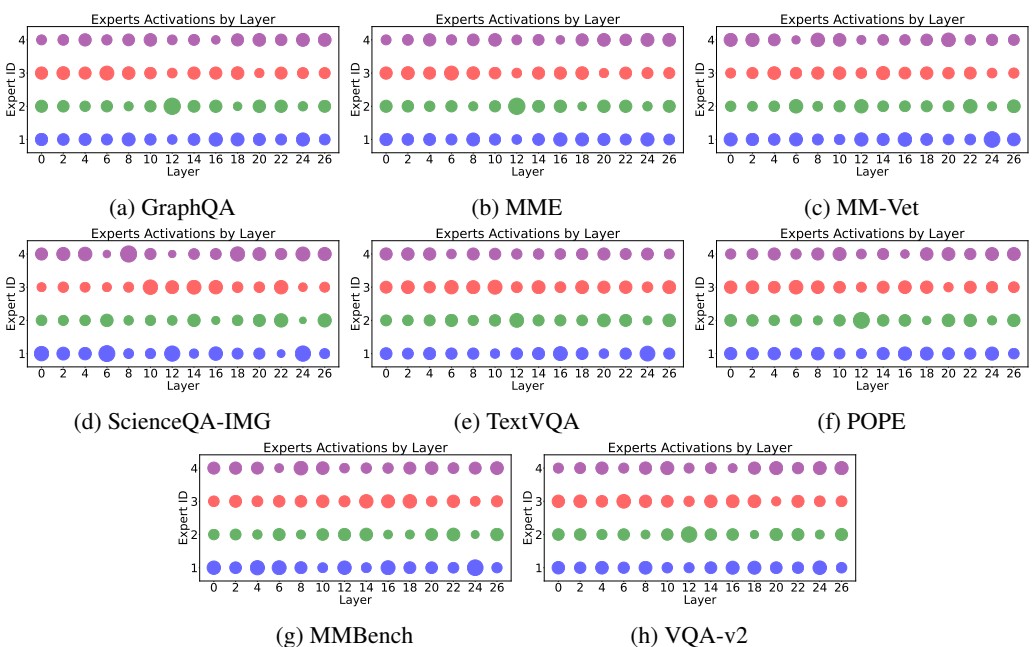

Figure 9: Layer-wise expert activations on DARE (Qwen2-1.5B). Bubble sizes are nearly uniform across layers, indicating balanced expert utilization.

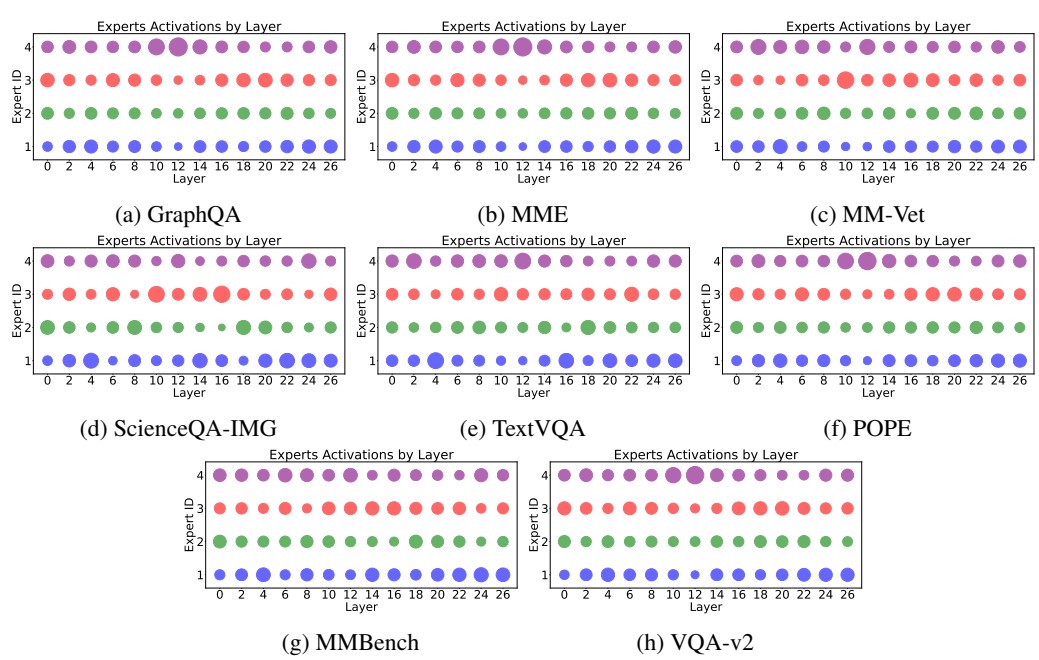

Figure 10: Layer-wise expert activations on DARE (Qwen3-1.7B). Bubble sizes are nearly uniform across layers, indicating balanced expert utilization.

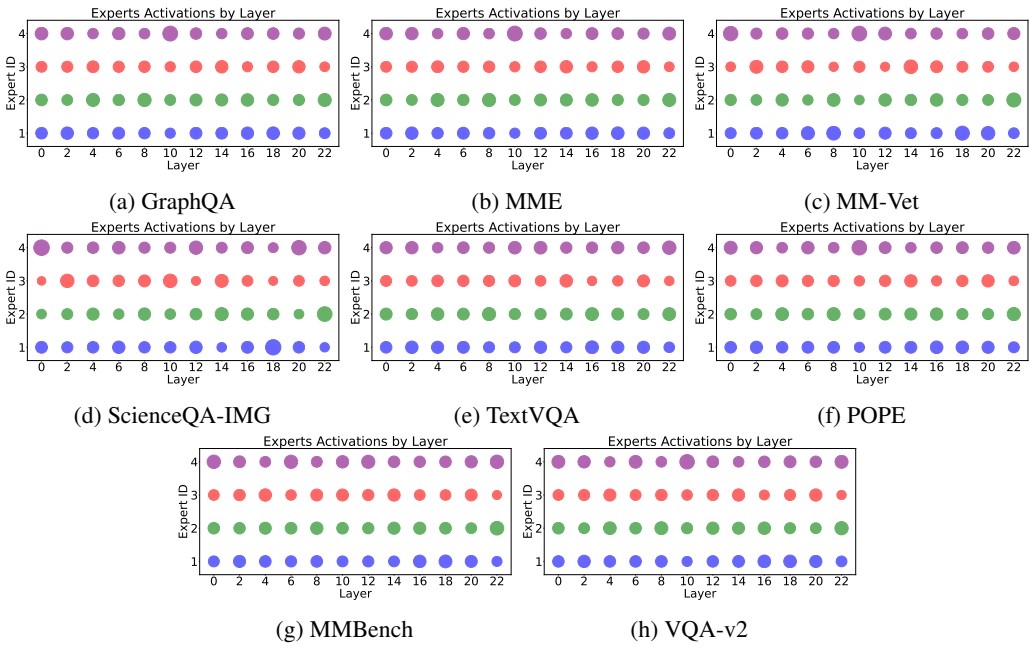

Figure 11: Layer-wise expert activations on DARE(Stablelm-1.6B). Bubble sizes are nearly uniform across layers, indicating balanced expert utilization.

close to zero, indicating that the experts learn highly distinct and complementary representations rather than collapsing to redundant behaviors. This observation confirms that the routing mechanism encourages specialization and maintains diversity of expertise throughout the network.

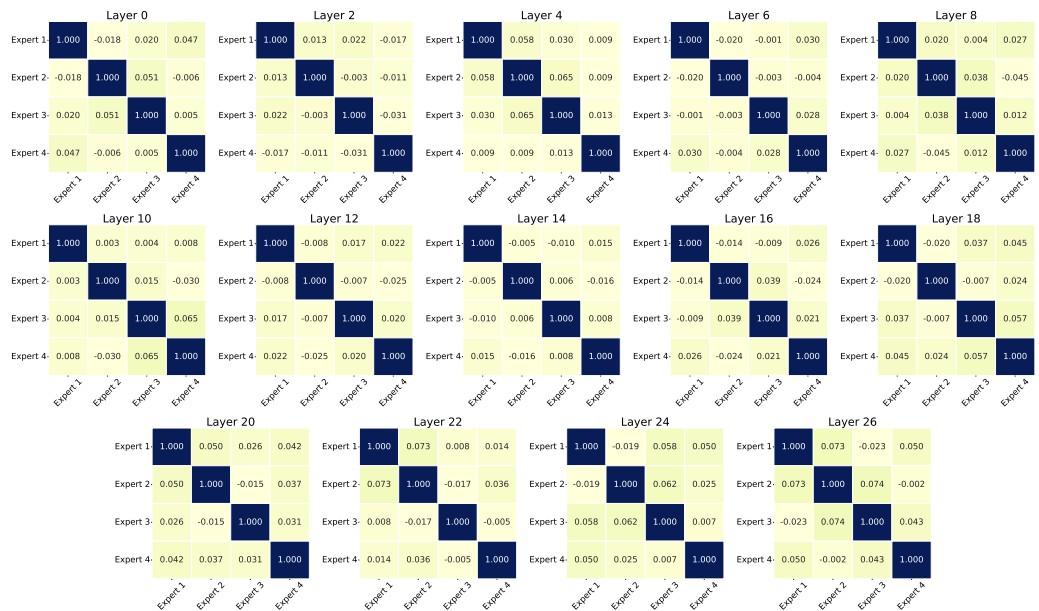

Figure 12: Layer-wise expert similarity matrix on DARE(Qwen2-1.5B). Cosine similarities between experts are measured at test time for each layer. Across all layers, inter-expert similarities remain near zero, indicating that the experts learn largely distinct representations

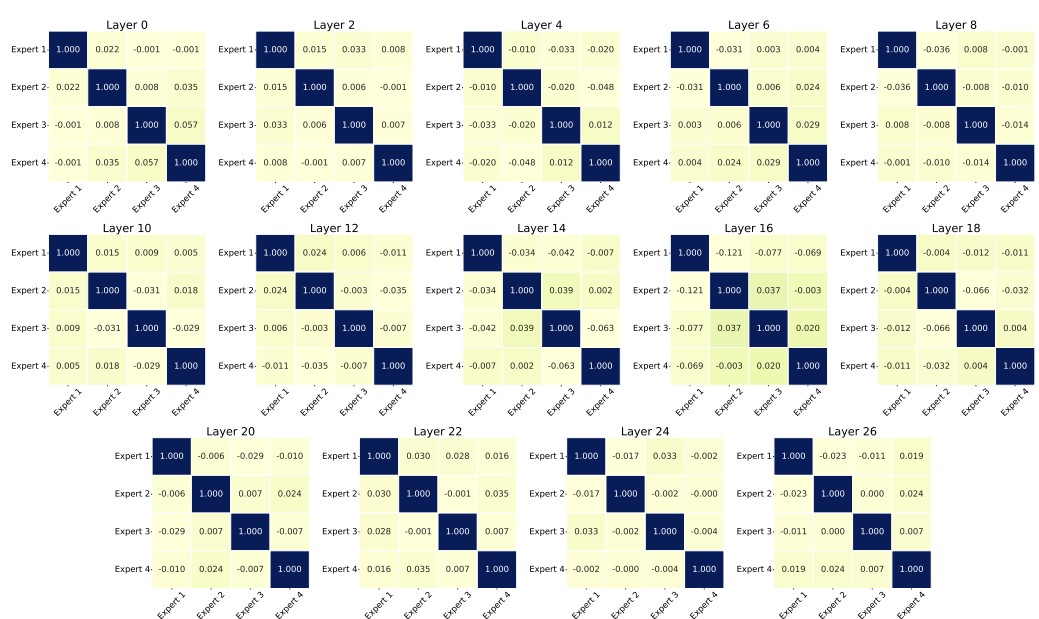

Figure 13: Layer-wise expert similarity matrix on DARE(Qwen3-1.7B). Cosine similarities between experts are measured at test time for each layer. Across all layers, inter-expert similarities remain near zero, indicating that the experts learn largely distinct representations

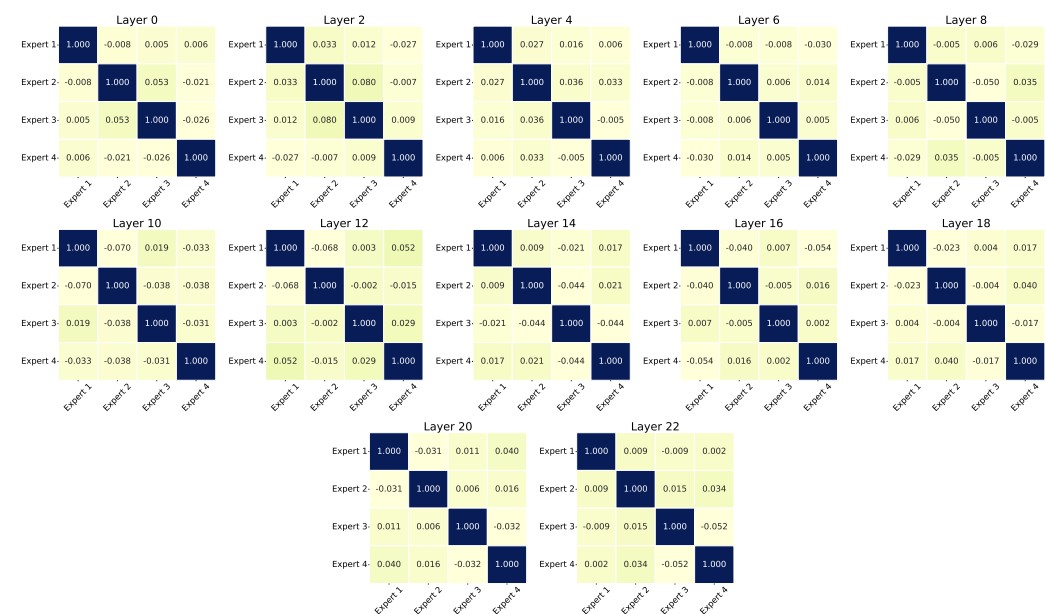

Figure 14: Layer-wise expert similarity matrix on DARE(StableLM-1.6B). Cosine similarities between experts are measured at test time for each layer. Across all layers, inter-expert similarities remain near zero, indicating that the experts learn largely distinct representations.

## F.4 CASE STUDY

To further illustrate the effectiveness of our approach, we present several representative case studies in Table 10. These examples highlight the superior routing behavior and improved task performance enabled by DARE.

Table 10: Exhibition Board of DARE(Qwen2-1.5B). DARE demonstrates the ability to detect and answer challenging questions when prompted to verify them.

**User:** Where is the flower in the image?

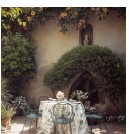

**DARE:** The flower is located on the table in the image.

**User:** What is the name of the man in the image?

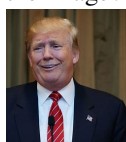

**DARE:** The man in the image is Donald Trump, the 45th President of the United States.

**User:** Do you know who drew this paintings?

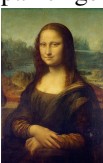

**DARE:** Yes, the painting of the woman with long hair and a big smile is by the famous artist Leonardo da Vinci.

**User:** What is the building in the image?

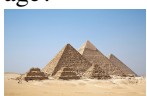

**DARE:** The building in the image is a pyramid, which is a large, flat structure built on a desert plain.

**User:** What is the fruit in the image?

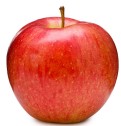

**DARE:** The fruit in the image is an apple.

**User:** What is unusual about this image?

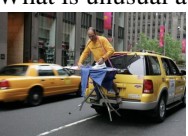

**DARE:** The unusual aspect of this image is that a man is ironing clothes using an ironing board while standing in the back of a taxi cab. This is not a typical scene, as one would expect to see someone ironing clothes in a more common indoor setting, such as a home or a laundry room. The man's decision to iron clothes in the back of a taxi cab adds an element of surprise and humor to the scene.

**User:** Describe the image and where's the red arrow?

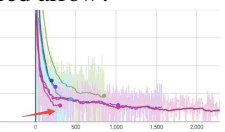

**DARE:** The image is a graph showing a line of data, with a red arrow pointing to the left. The arrow is located at the bottom of the graph, indicating a direction or a change in the data.

**User:** Can you give a short introduction to this movie?

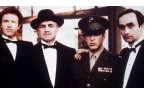

**DARE:** The movie is called "The Godfather," and it is a classic crime drama film directed by Francis Ford Coppola. The film is set in the 1940s and 1950s and features a powerful cast, including Al Pacino, Robert De Niro, and Joe Pesci. The movie tells the story of the Corleone family, led by patriarch Vito Corleone, as they navigate the world of organized crime and deal with various challenges and conflicts. The film is known for its intense action, compelling characters, and memorable performances, making it a classic and influential piece of cinema.

