# OpenReview forum: "DARE:Difficulty-Aware Dynamic Routing for Mixture of Experts"
_ICLR.cc/2026/Conference — ICLR 2026 Conference Withdrawn Submission_

### Official Review · Reviewer_bBMU · 2025-10-30

**Soundness:** 1
**Presentation:** 3
**Contribution:** 2
**Rating:** 4
**Confidence:** 4

**Summary:**

This paper proposes DARE, a dynamic routing mechanism for MoE architectures that explicitly models token-level difficulty when allocating experts. Instead of using a fixed number of experts per token (as in Top-K routing), DARE introduces a lightweight difficulty predictor based on log-perplexity to estimate the complexity of each token. A learnable thresholding mechanism then dynamically determines how many experts to activate per token, enabling adaptive and efficient resource allocation. Experiments show that DARE improves accuracy while reducing computation by 39% compared to Top-K routing, outperforming recent dynamic methods.

**Strengths:**

1. Dynamic routing is a core topic in the study of MoE routing mechanisms.

2. Intuitively, the dynamic routing design of this paper makes sense.

**Weaknesses:**

1. _Overclaim_. Some prior works have also designed dynamic routing strategies based on token-level difficulty, such as Top-p and DYNMoE. Therefore, Lines 61–62 appear somewhat overclaimed. In addition, although the authors provide empirical comparisons with existing dynamic routing approaches, they do not conceptually clarify the advantages of the proposed prediction-based method over these approaches, raising concerns about its generalization.

2. _Unclear overhead of the difficulty predictor_. The token difficulty is estimated through an additional difficulty predictor. However, the paper does not report the computational overhead introduced by the difficulty predictor.

3. _Insufficient experiments_. The experiments are mainly conducted on MoE-LLaVA, without validation on other MoE models. Moreover, MoE-LLaVA (2023) is not a native MoE model but a dense model upcycled into an MoE variant, and its sparsity ratio (activated experts / total experts) is relatively high, which deviates considerably from current mainstream MoE configurations. Thus, further experiments on recent and representative MoE architectures are needed to validate the effectiveness of the proposed method.

4. The proposed method does not seem to include any design tailored for multimodal scenarios, so it would be helpful to add experiments on language-only tasks.

In conclusion, this work lacks a clear conceptual comparison with existing difficulty-based dynamic routing methods, making it difficult to assess the effectiveness of the proposed approach across different scenarios. Furthermore, the absence of experiments on recent mainstream MoE models further raises concerns about the effectiveness. Therefore, my current rating is Reject.

**Questions:**

Please refer to Weaknesses.

---

> ### Author Response · Authors · 2025-11-20
> **Response to reviewer bBMU (Part 1)**
>
> We appreciate the reviewer’s comments and provide detailed clarifications to address the concerns below.
>
> > Weaknesses 1: Overclaim
>
> We clarify that although both **Top-p** and **DynMoE** introduce adaptive expert selection, neither explicitly models *token-level difficulty* as defined in our work.
>
> Specifically, **Top-p routing** relies on the entropy of the gating distribution to decide the number of activated experts. Gating entropy measures the *router's uncertainty* about expert choice rather than the *intrinsic difficulty* of processing a token, and thus does not consistently correlate with token-level loss. Moreover, Top-p only employs this uncertainty *implicitly* through probabilistic truncation, without defining difficulty as a learnable quantity.
>
>
>
> In contrast, **DynMoE** determines expert activation by comparing the similarity between each token embedding and expert embedding against a learned threshold. This heuristic rule reflects *affinity to experts* rather than *how difficult the token itself is to process*. It lacks an explicit formulation of difficulty and cannot distinguish between easy and hard tokens that happen to have similar embeddings.
>
> Our method **DARE** is the first to introduce an *explicit and interpretable difficulty proxy*, namely log-perplexity, which is directly derived from the model's predictive behavior. This formulation provides a principled and measurable signal that guides adaptive expert allocation and clearly distinguishes DARE from previous uncertainty-based or similarity-based routing strategies.
>
> > Weakness 2: Unclear overhead of the difficulty predictor.
>
> We clarify that all reported efficiency metrics in the paper, including the number of activated parameters and inference cost, already include the cost of the difficulty predictor.
>
> The predictor is implemented as a 2-layer MLP that operates on the token hidden states and contains only a negligible number of parameters compared to the overall model size. As shown in Table 1, its contribution to the total parameter count and FLOPs is below 0.01%, which is within the noise margin of measurement. The following table reports the exact parameter scale on the Qwen2-1.5B backbone.
>
> **Table 1.** Parameter and computational overhead of the difficulty predictor relative to the main model.
>
> | Component | Parameters | Ratio to Total (%) | GFLOPs/token| Ratio to Total (%) |
> |------------|----------------|--------------------|--------------------|--------------------|
> | Main Model | 3.72B | 100.0 | 64.7| 100.0 |
> | Single Difficulty Predictor | 0.4M | 0.011 |0.0055|  0.0085 |
>
> Given this minimal scale, the difficulty predictor introduces no measurable slowdown or memory overhead in practice. Thus, all efficiency comparisons in the main text already reflect the full cost of DARE.

---

> > ### Author Response · Authors · 2025-11-20
> > **Response to reviewer bBMU (Part 2)**
> >
> > > Weakness 3: Insufficient experiments.
> >
> > We would like to clarify that **MoE-LLaVA** is currently the most widely adopted framework for studying *multimodal* Mixture-of-Experts architectures in the academic community. It provides a standard and well-recognized testbed for analyzing routing mechanisms under realistic multimodal settings.
> >
> > While several recent multimodal MoE systems have explored larger-scale or proprietary implementations, most existing open-source or research-accessible models follow the MoE-LLaVA design paradigm. In contrast, *native* multimodal MoE models typically contain over 30 billion parameters, making large-scale retraining or ablation studies infeasible in academic environments.
> >
> > Therefore, MoE-LLaVA serves as a computationally practical and representative platform for evaluating routing strategies such as DARE. Combined with our additional results on multiple backbones (Qwen2-1.5B, Qwen3-1.7B, and StableLM-1.6B), we believe this provides sufficient evidence of the method's generality and robustness across architectures.
> >
> > > Weakness 4: Add experiments on language-only tasks.
> >
> > We thank the reviewer for the valuable suggestion to evaluate DARE on a language-only model. Due to time and computational resource constraints, full pre-training was not feasible. To further assess the generality of our method beyond multimodal MoE settings, we conducted additional experiments on the **Qwen2-0.5B** architecture. We set the number of experts to 16 and performed instruction fine-tuning on the OpenR1-Math-220k dataset, following the same routing comparison (Top-2 vs. DARE), we evaluated the models on the **HellaSwag**, **ANLI1,2,3** and **BoolQ** benchmarks.
> >
> > **Table 2.** Performance of DARE on a language-only MoE model (Qwen2-0.5B × 16 experts).
> >
> > | Model | Avg-K ↓ |HellaSwag ↑ | ANLI-1 ↑ | ANLI-2 ↑ | ANLI-3 ↑ | BoolQ |
> > |:------|:----------------:|:-------------:|:-------------:|:----------------:|:----------------:|:----------------:|
> > | Qwen2-MoE (Top-2) |2.00 |0.3699 |0.3330 | 0.3330 |  0.3350 | 0.6217 |
> > | Qwen2-MoE (DARE, ours) | **1.61** |**0.3703**| 0.3330| 0.3330 |  0.3350 | 0.6217 |
> >
> > The results in Table 2 demonstrate that DARE maintains comparable or slightly better performance while the average number of activated experts is reduced. These findings confirm that the proposed difficulty-aware dynamic routing mechanism generalizes effectively to **language-only MoE architectures**, reinforcing its conceptual and practical generality.

---

> ### Author Response · Authors · 2025-11-27
> **Looking Forward to Your Response**
>
> Dear Reviewer bBMU,
>
> We greatly appreciate the time you've invested in reviewing our response. Having submitted our rebuttal, we are eager to know if our response has addressed your concern. The discussion phase is nearing its end, but we have not yet received your response. We look forward to hearing from you for any further clarification that you might require.
>
> Best,
>
> Submission 8748 Authors.

---

### Official Review · Reviewer_Gk9W · 2025-10-31

**Soundness:** 3
**Presentation:** 3
**Contribution:** 3
**Rating:** 6
**Confidence:** 4

**Summary:**

This paper introduces DARE, a novel dynamic routing strategy for Mixture-of-Experts (MoE) models in the context of Large Vision-Language Models (LVLMs). The core idea is to dynamically allocate a variable number of experts to each token based on its estimated difficulty, which is proxied by log-perplexity. By introducing a lightweight difficulty predictor and an online threshold adaptation mechanism, DARE aims to assign more computational resources to complex tokens and fewer to simple ones, thereby improving both task performance and computational efficiency compared to static Top-K routing and other existing dynamic methods.

**Strengths:**

1. This paper is well-motivated. It addresses a significant and well-recognized limitation of standard MoE architectures. DARE chooses the logarithmic perplexity as a proxy for difficulty, which is a theoretically well-grounded metric directly related to the model’s own uncertainty and loss function.
2. This paper conduct extensive experiments across multiple standard vision-language benchmarks and on various LLM backbones.
3. The paper is well-written, logically structured, and easy to follow.

**Weaknesses:**

1. Dependence on the predefined target distribution \Pi. The online threshold adaptation mechanism relies on a manually predefined target expert assignment distribution \Pi (for example, for Qwen2-1.5B, Π = [0.6, 0.3, 0.09, 0.01]). This makes \Pi a critical, model-specific hyperparameter.
2. Limitation of the Difficulty Proxy: The paper relies solely on log-perplexity as the difficulty measure. While this effectively captures prediction uncertainty, it does not reflect semantic complexity or task relevance. Some low-perplexity tokens may still require deeper reasoning, while high-perplexity tokens could simply be rare but unimportant. The paper would benefit from discussing these limitations and considering alternative or combined proxies—such as integrating perplexity with gradient-based importance or attention scores.
3. The Number of Experts. All experiments in this paper appear to be conducted with only four experts per MoE layer (M = 4). It remains an open and important question how DARE performs and behaves when the number of experts increases. Would using a simple quantile-based threshold still work well, or could it cause some experts to be used less than they should? A discussion or empirical analysis on the scalability of DARE under larger MoE configurations would make the paper more convincing.

**Questions:**

1. Regarding the target distribution \Pi: Have you experimented with learning this distribution or using a parametric form instead of setting discrete values manually? How sensitive are the results to small perturbations in the \Pi_k values? Providing some intuition on how to set \Pi for a new model would be very helpful for practitioners.
2. Is the architecture of the difficulty predictor (a simple MLP) sufficient? Have you tried more complex structures, such as using attention mechanisms to aggregate contextual information for more accurate difficulty prediction?
3. In the Analysis of Expert Load Balance(Figure 4), DARE demonstrates outstanding performance. Does this balance come at the expense of expert specialization? Could a more quantitative analysis be provided—for example, measuring each expert’s activation preferences when processing different types of tokens (such as nouns, verbs, or domain-specific terms)? This would offer deeper insights into expert behavior under the DARE routing mechanism.

---

> ### Author Response · Authors · 2025-11-20
> **Response to reviewer  Gk9W (Part 1)**
>
> We sincerely appreciate your acknowledgment of our work and your recognition of our method. Regarding your concerns, we hope that our clarification can be helpful.
>
> > Weaknesses 1 & Question 1: Feasibility of Learning $\Pi$
>
> We thank the reviewer for the insightful suggestion regarding learning the target distribution $\Pi$. We indeed explored making $\Pi$ learnable instead of predefined. However, this approach introduces major technical challenges: the thresholds $\tau_j$ are computed from the quantiles of $\Pi$, which are non-differentiable, preventing gradient-based optimization of $\Pi$ or its parametric form.
>
> To overcome this, we experimented with a **soft relaxation** using a sigmoid-based approximation to make the quantile operation differentiable. As shown in Table 1, this approach led to highly unstable training and inferior performance, likely because the softened thresholds fluctuate excessively, hindering stable learning of the routing behavior.
>
> **Table 1.** Comparison between fixed $\Pi$ and learnable (soft-relaxed) $\Pi$ on Qwen2-1.5B × 4 experts.
>
> | $\Pi$ Type | MME ↑ | MMB ↑ | TextVQA ↑ | GQA ↑ |
> |:--------|------:|------:|------:|-------------:|
> | Learnable (soft-relaxed) |1364.5  | 64.6  | 55.2  | 61.9       |
> | Fixed (ours) | **1404.1** | **66.1** | **57.0** | **62.0** |
>
> These results motivated our final **Online Threshold Adaptation** design, which keeps $\Pi$ fixed but dynamically tracks the empirical difficulty distribution via EMA updates. This achieves stable and robust adaptation without introducing non-differentiable or unstable components.
>
> > Weaknesses 3: Limited Expert Count
>
> We agree that evaluating only 4 experts may limit the generality of the conclusions. To verify DARE's scalability, we trained an additional model using **Qwen2-0.5B with 16 experts** under the same setup. As shown in Table 5, DARE maintains consistent advantages over fixed Top-K routing: performance remains better almost across all benchmarks while the average number of activated experts is substantially reduced.  This confirms that the proposed difficulty-aware routing generalizes beyond the 4-expert configuration.
>
> **Table 5.** Comparison between Top-2 and DARE on Qwen2-0.5B × 16 experts. DARE preserves accuracy while improving efficiency.
>
> | Method          | #Experts | Avg-K ↓ |    MME ↑ |    MMB ↑  |    GQA ↑ |    SQA ↑ |  TextVQA ↓ |   POPE ↑   |   MMVet ↑ |
> |---------|----------:|--------:|------:|------:|------:|-------------:|---------:|--------------:|--------------:|
> | Top-2 MoE       | 16 | 2.00     | 1198.66     | 53.87     | 57.84    | 59.35    | 45.81      | 86.78     |   25.10     |
> | **DARE (ours)** | 16 | **1.56** | **1267.64** | **56.36** | **59.33**| **61.48**| **47.82** | 86.34 |**26.40** |
>
> These results demonstrate that DARE scales effectively with larger expert pools, validating that its advantage is not tied to the 4-expert setting.
>
>
> > Question 2: Difficulty Predictor MLP Sufficiency
>
> The difficulty prediction module is formulated as a **scalar regression task**, where the goal is to estimate a continuous token-level difficulty score (log-perplexity) from hidden representations. Given this formulation, a small MLP is a natural and sufficient choice, it introduces minimal computational overhead while effectively capturing nonlinear relationships.
>
> Incorporating attention-based predictors (e.g., Transformer layers) would substantially increase computation and latency, which conflicts with DARE's objective of improving routing efficiency. While exploring more expressive predictors could be an interesting direction for future work, our current lightweight design achieves accurate estimation with negligible additional cost (~0.1% of total params), aligning with DARE's efficiency-oriented motivation.
>
> > Question 3: Load Balance vs. Expert Specialization
>
> We appreciate the reviewer's interest in the specialization behavior of experts under DARE. As shown in **Figures 12–14** of appendix in the paper, we visualize the pairwise cosine similarities between expert representation vectors across multiple layers and datasets. The results show that most expert pairs remain **nearly orthogonal**, indicating that DARE preserves diverse and specialized expert representations even while improving load balance.
>
> This analysis suggests that the enhanced balance observed in Figure 4 does not come at the cost of expert specialization. Instead, DARE encourages a more efficient utilization of experts without collapsing them into redundant functions.

---

> > ### Author Response · Authors · 2025-11-20
> > **Response to reviewer Gk9W (Part 2)**
> >
> > > Weakness 2: Log-Perplexity Limitations and Alternative Proxies
> >
> > While we agree that perplexity primarily reflects prediction uncertainty, our analysis indicates that it also implicitly captures **semantic and contextual complexity**.
> >
> > To illustrate this, we computed token-level perplexities of several frequent tokens under both *simple* (easy perception) and *complex* (reasoning) question–answer pairs using the Top-2 MoE baseline. As shown in Table 2, the same tokens exhibit consistently higher perplexity in complex tasks, indicating that the measure correlates with contextual difficulty rather than mere frequency or rarity.
> >
> > **Table 2.** Average token-level log-perplexity (Top-2 MoE) under simple vs. complex questions.
> >
> > | Token | Simple QA ↓ | Complex QA ↓ | Δ (Complex – Simple) ↑ |
> > |:------|-------------:|-------------:|----------------------:|
> > | "red" |  0.0609       | 4.2930       | +4.2321              |
> > | "is"  | 0.1135        | 0.3123       | +0.1988              |
> > |"apple"|  0.0069       | 0.0178       | +0.0109              |
> >
> > This suggests that log-perplexity reflects not only model uncertainty but also token usage difficulty conditioned on task semantics.
> >
> > And then we discuss why other potential indicators such as gating entropy, gradient norms, and attention based metrics are suboptimal in this context.
> >
> > #### Gating Entropy.
> > Entropy of the routing distribution quantifies the uncertainty of the router over expert selection rather than the intrinsic modeling difficulty of a token. High entropy simply indicates that the router assigns similar probabilities to multiple experts, not how hard a token is to predict. Beyond this theoretical mismatch, we also empirically find that entropy is a poor proxy because its distribution shows very low variance. To demonstrate this, we computed the gating entropy statistics of top-2 routing across three vision language benchmarks. As shown in Table 3, the entropy exhibits very limited variation across datasets (standard deviation $\approx$ 0.06), indicating that it fails to meaningfully differentiate token difficulty. In contrast, log perplexity shows a clearly long-tailed distribution that aligns with token-level loss, making it a more faithful and discriminative difficulty proxy.
> >
> > **Table 3.** Gating entropy statistics across benchmarks. Entropy values exhibit very limited variation (Std $\approx$  0.06), indicating weak discrimination of token difficulty.
> >
> > | Dataset     | Number of sample | Number of tokens | Gating Entropy per token Mean | Std   |
> > |------------:|----------------:|----------------:|------------------------------:|------:|
> > | MME        | 33236            | 26199712         | 1.345                         | 0.057 |
> > | TextVQA    | 70000            | 58327500         | 1.342                         | 0.061 |
> > | GQA        | 176092           | 138376462        | 1.346                         | 0.058 |
> >
> > We further performed a controlled experiment where gating entropy was used as the difficulty signal in place of log perplexity. As shown in Table 4, this substitution leads to a consistent performance drop across benchmarks, confirming that entropy is not an effective proxy for token difficulty.
> >
> > **Table 4.** Comparison between log perplexity and gating entropy as difficulty signals. Using entropy results in consistent degradation in accuracy and efficiency.
> > | Difficulty Signal     | MME↑ | MMB↑ | TextVQA↑ | SQA↑ |
> > |------------------|------:|------:|------:|-------------:|
> > | Gating Entropy           | 1352.6  | 65.1  | 55.5  | 69.2       |
> > | Log Perplexity (ours)    |**1404.1** | **66.1** | **57.0** | **69.9** |
> >
> > #### Gradient Norms.
> >
> > Gradient-based difficulty estimation measures a token’s influence on parameter updates and is conceptually meaningful, but its dependence on per-token gradient norms makes it computationally infeasible. The required non-aggregated gradients introduce prohibitive costs in two aspects, exacerbated by the scale of MoE models. Loss of computational efficiency: Standard backpropagation is efficient because it aggregates per-token gradients into a single tensor and leverages highly optimized GEMM operations. Computing individual per-token gradients bypasses this mechanism, causing substantial overhead and poor GPU utilization. Prohibitive memory footprint: Per-token gradient computation also requires storing tensors proportional to batch size × total parameters, resulting in memory demands far exceeding typical hardware capacity. Owing to these extreme compute and memory costs, gradient norms are not a practical supervisory signal in our setting.
> >
> > #### Attention based Metrics.
> > Attention statistics such as attention entropy or sparsity describe how tokens aggregate contextual information but do not directly quantify how difficult they are to model. Their relationship to predictive difficulty is indirect and inconsistent across architectures. Consequently, they fail to provide a stable signal for expert allocation.

---

> ### Author Response · Authors · 2025-11-27
> **Looking Forward to Your Response**
>
> Dear Reviewer Gk9W,
>
> We greatly appreciate the time you've invested in reviewing our response. Having submitted our rebuttal, we are eager to know if our response has addressed your concern. The discussion phase is nearing its end, but we have not yet received your response. We look forward to hearing from you for any further clarification that you might require.
>
> Best,
>
> Submission 8748 Authors.

---

### Official Review · Reviewer_VRoW · 2025-11-01

**Soundness:** 2
**Presentation:** 3
**Contribution:** 2
**Rating:** 4
**Confidence:** 4

**Summary:**

The paper focuses on routing strategies of sparse MoE in vision-language models, noting that a fixed Top-K causes excessive computation for easy tokens and insufficient computation for difficult ones. It therefore proposes DARE: a lightweight MLP first predicts token difficulty from hidden representations (using log perplexity as an interpretable proxy), then maps the predicted difficulty to “the number of experts to activate in this layer” through a set of learnable thresholds, thus dynamically deciding how many experts to use per token. The authors provide an analysis of the negative correlation between difficulty and accuracy, the complete training objective (task CE, difficulty regression, load balancing), and multi-benchmark experiments based on Qwen2-1.5B, Qwen3-1.7B, and StableLM-1.6B. The main conclusion is: on multiple benchmarks, performance is comparable to or better than strong baselines, while reducing the average number of activated experts to about 1.22, and achieving actual gains in inference FLOPs, throughput, and latency.

**Strengths:**

The most commendable aspect is the closed loop between motivation and evidence: from the correlation observation in Fig. 2, to using log perplexity for difficulty supervision, to mapping difficulty into “number of experts” through threshold stratification—the technical line is clear, interpretable, and consistently reproducible across benchmarks.
In terms of efficiency, the authors not only report a significant reduction in the average number of activated experts, but also demonstrate end-to-end benefits under the same hardware setup: FLOPs/token reduced to 52.25, throughput increased to 75 tok/s, latency shortened to 5.4 s, with memory usage almost unchanged compared to Top-2 MoE-LLaVA.
More detailed visualizations (e.g., layer–task activation counts, load-balancing differences across methods) further aid understanding of routing mechanisms.
It’s worth emphasizing your observation that “simply reducing activated experts can speed up inference with almost no performance drop, especially under fixed-expert + Top-K setups”—this paper’s data shows traces of that: compared to fixed Top-2, DARE’s lower Avg-K yields similar scores, and efficiency gains align with the drop in Avg-K. However, cases like MoE++ and DynMoE show performance degradation when Avg-K ≈ 1, suggesting that “reducing Avg-K” alone isn’t sufficient—difficulty-aware routing still makes a difference.

**Weaknesses:**

The biggest comparability issue lies in the “number of experts” setting. Using 4 experts may underestimate the potential of dynamic baselines, so conclusions drawn solely from 4-expert setups have limited persuasiveness. For concerns like “should expert-selection methods be tested with more experts,” I agree that replication should at least include the common scale 16 to avoid conclusions tied to low-expert special cases.

The efficiency evaluation chain is also incomplete. The speed comparison only directly measures DARE vs. MoE-LLaVA (Top-2) in memory, FLOPs, throughput, and latency; neither the main paper nor the appendix includes speed results for MoE++,Top-p, ReMoE, or DynMoE, which appear in the performance tables. There are also no controlled experiments under “same Avg-K” or “same number of activated parameters,” leaving the efficiency gain from “difficulty awareness” mixed with that from “fewer activated experts.” Your third comment fully applies here—those baselines should be added with Avg-K alignment.

Statistical robustness still needs improvement. The main result table lacks error bars or repetitions (the efficiency table mentions “means over five independent runs,” but performance tables do not), making a few-to-ten-point gap hard to judge for significance.

**Questions:**

Please include additional experiments with multiple experts and improve the speed tests. A more thorough response and supplementation of these issues may positively influence my assessment of the paper’s persuasiveness.

---

> ### Author Response · Authors · 2025-11-20
> **Response to reviewer  VRoW**
>
> We thank the reviewer for their rigorous and thought-provoking feedback. We address the points raised below.
>
> > Weaknesses 1: Limited Expert Count
>
> We agree that evaluating only 4 experts may limit the generality of the conclusions. To verify DARE's scalability, we trained an additional model using **Qwen2-0.5B with 16 experts** under the same setup. As shown in Table 1, DARE maintains consistent advantages over fixed Top-K routing: performance remains better almost across all benchmarks while the average number of activated experts is substantially reduced. This confirms that the proposed difficulty-aware routing generalizes beyond the 4-expert configuration.
>
> **Table 1.** Comparison between Top-2 and DARE on Qwen2-0.5B × 16 experts. DARE preserves accuracy while improving efficiency.
>
> | Method          | #Experts | Avg-K ↓ |    MME ↑ |    MMB ↑  |    GQA ↑ |    SQA ↑ |  TextVQA ↓ |   POPE ↑   |   MMVet ↑ |
> |---------|----------:|--------:|------:|------:|------:|-------------:|---------:|--------------:|--------------:|
> | Top-2 MoE       | 16 | 2.00     | 1198.66     | 53.87     | 57.84    | 59.35    | 45.81      | 86.78     |   25.10     |
> | **DARE (ours)** | 16 | **1.56** | **1267.64** | **56.36** | **59.33**| **61.48**| **47.82** | 86.34 |**26.40** |
>
> These results demonstrate that DARE scales effectively with larger expert pools, validating that its advantage is not tied to the 4-expert setting.
>
> > Weaknesses 2: Incomplete Efficiency Baselines and Avg-K Alignment Feasibility
>
> We thank the reviewer for pointing out that our efficiency evaluation previously included only Top-2 MoE-LLaVA. To ensure a fair comparison, we additionally measured **GFLOPs/token** for dynamic routing baselines under the same hardware. The results are summarized in Table 2.
>
> **Table 2.** Inference efficiency comparison across routing strategies (Qwen2-1.5B × 4 experts).
>
> | Method | Avg-K ↓ | GFLOPs / token ↓ |
> |:--------|--------:|----------------:|
> | Top-2 MoE | 2.00 | 76.9|
> | MoE++ | 0.90∗ |  69.9|
> | ReMoE | 1.59 | 66.0|
> | DynMoE | 1.00 | 59.2|
> | **DARE (ours)** | 1.22 | 64.8|
>
> These results show that DARE is highly competitive in terms of computational efficiency, while operating with a reduced average activation budget.
>
> **Clarification on the "Controlled Experiment" at "Same Avg-K"**:
> We would like to address the reviewer's concern regarding the "confounding variable."
>
> Avg-K is an **Emergent Property**, not a **Hyperparameter**: for most dynamic routing methods (including DARE, ReMoE, DynMoE, and Top-p), Avg-K, the average number of activated experts, is not a tunable hyperparameter. Rather, it emerges from the interaction between the learned routing policy and the data.
>
> **Difficulty-Awareness $\rightarrow$ Fewer experts  $\rightarrow$  Efficiency**: while it is true that efficiency gains arise from activating fewer experts, the key question is how to reduce Avg-K without degrading performance. DARE demonstrates that "difficulty-awareness" provides a more effective mechanism to achieve this.
>
> In short, a controlled "same Avg-K" comparison is not feasible, as it would require fundamentally altering the learned routing policies under evaluation.
>
> > Weaknesses 3: Statistical Robustness
>
> We appreciate the reviewer's comment on statistical robustness. To address this, we conducted three independent runs for several primary benchmark using random seeds and report the mean and standard deviation in Table 3. The results confirm that DARE's improvements are consistent across runs, indicating stable optimization and statistically reliable performance.
>
> **Table 3.** Main results averaged over three random seeds (Qwen2-1.5B × 4 experts).
>
> | Method       | MME ↑ | MMB ↑ | SQA ↑ | TextVQA ↑ |
> |-------------|------:|------:|------:|-------------:|
> | Top-2 MoE    | 1368.2 $\pm$ 4.7 | 64.8 $\pm$ 0.2 | 69.73 $\pm$ 0.06 | 56.76  $\pm$ 0.03     |
> | **DARE**     | **1404.2** $\pm$ 0.0 | **66.1** $\pm$ 0.0| **69.9**$\pm$ 0.0 | **57.0**$\pm$ 0.0 |
>
>
> As shown in Table 3, DARE consistently outperforms the Top-2 baseline. A notable observation is DARE's zero variance across runs ($\pm$ 0.0). We clarify that the performance fluctuation observed in the standard Top-2 MoE baseline stems from its design: Top-2 typically introduces explicit noise injection into the router logits which inherently results in performance variance across different random seeds. In contrast, dynamic routing algorithms (such as DARE, ReMoE, and DynMoE) do not utilize this explicit noise injection mechanism. Consequently, when the generation temperature is set to 0 (our evaluate setting), DARE (and other similar dynamic methods) exhibits stability and ensures a perfectly stable and reproducible routing path across runs.

---

> ### Author Response · Authors · 2025-11-27
> **Looking Forward to Your Response**
>
> Dear Reviewer VRoW,
>
> We greatly appreciate the time you've invested in reviewing our response. Having submitted our rebuttal, we are eager to know if our response has addressed your concern. The discussion phase is nearing its end, but we have not yet received your response. We look forward to hearing from you for any further clarification that you might require.
>
> Best,
>
> Submission 8748 Authors.

---

### Official Review · Reviewer_f5dZ · 2025-11-10

**Soundness:** 2
**Presentation:** 3
**Contribution:** 2
**Rating:** 4
**Confidence:** 4

**Summary:**

This paper introduces DARE (Difficulty-Aware Dynamic Routing), a novel routing mechanism for Sparse Mixture-of-Experts (MoE) models. The central thesis is that the conventional Top-K routing strategy is suboptimal as it allocates a fixed computational budget (i.e., a fixed number of experts) to every token, irrespective of its intrinsic complexity. To address this, DARE proposes a dynamic allocation strategy guided by an explicit measure of token-level difficulty. Specifically, the method uses log-perplexity as a proxy for difficulty. A lightweight MLP, termed the "difficulty predictor," is trained alongside the main model to estimate this difficulty score for each token's hidden representation. Based on the predicted score, a variable number of experts $K^*$ is selected for each token by comparing the score against a set of dynamically adjusted thresholds. These thresholds are updated online using an exponential moving average (EMA) on the quantiles of the difficulty distribution within each mini-batch, aiming to match a predefined target distribution of expert counts. The authors evaluate DARE on several vision-language benchmarks, demonstrating superior performance and computational efficiency compared to fixed Top-K routing and other recent dynamic routing strategies.

**Strengths:**

1.  The paper correctly identifies a key limitation of the standard Top-K routing in MoE models and provides a clear motivation for a more adaptive approach.
2.  Despite methodological concerns, the paper presents compelling empirical results across multiple vision-language benchmarks and three different model backbones. DARE consistently outperforms various baselines, including both static and dynamic routing methods, often while using fewer computational resources.
3. The authors provide a good set of analyses, including ablations on the core components of their method, visualizations of expert allocation patterns, and measurements of inference efficiency.

**Weaknesses:**

1.  No other potential difficulty signals (e.g., gating entropy, gradient norms, attention-based metrics) are explored or even discussed as alternatives in the main text. This makes the design choice seem arbitrary rather than principled.
2.  DARE introduces a non-trivial amount of new machinery: an MLP predictor with its own learning rate, an auxiliary loss weighted by $\alpha$, a target distribution $\Pi$ that must be specified per model, and an EMA momentum $\gamma$. This complexity is a clear disadvantage compared to the elegance of Top-K routing. The lack of a sensitivity analysis for these new hyperparameters is a glaring omission.

**Questions:**

1.  Could you provide a more detailed justification for using a separate, supervised difficulty predictor MLP instead of a simpler, unsupervised signal derived directly from the gating network's output distribution (e.g., its entropy)? Did you perform any experiments comparing these alternatives?
2.  The target distribution $\Pi$ appears to be a critical, manually-tuned hyperparameter. How were the specific values for each backbone model (e.g., $\Pi = (0.6, 0.3, 0.09, 0.01)$ for Qwen2-1.5B) determined? Please provide an ablation study showing how performance changes with different settings of $\Pi$, for instance, comparing your chosen "long-tailed" distribution to a uniform one and another differently-skewed one.

---

> ### Author Response · Authors · 2025-11-20
> **Response to reviewer f5dZ (Part 1)**
>
> We thank the reviewer for the thorough review of our work. We have taken note of the concerns raised by the reviewer and below we will address them accordingly.
>
> > Weaknesses 1 & Question 1: Unprincipled Signal Selection
>
> We clarify our rationale for adopting log perplexity as the token level difficulty signal and discuss why other potential indicators such as gating entropy, gradient norms, and attention based metrics are suboptimal in this context.
>
> #### Gating Entropy.
> Entropy of the routing distribution quantifies the uncertainty of the router over expert selection rather than the intrinsic modeling difficulty of a token. High entropy simply indicates that the router assigns similar probabilities to multiple experts, not how hard a token is to predict. Beyond this theoretical mismatch, we also empirically find that entropy is a poor proxy because its distribution shows very low variance. To demonstrate this, we computed the gating entropy statistics of top-2 routing across three vision language benchmarks. As shown in Table 1, the entropy exhibits very limited variation across datasets (standard deviation $\approx$ 0.06), indicating that it fails to meaningfully differentiate token difficulty. In contrast, log perplexity shows a clearly long-tailed distribution that aligns with token-level loss, making it a more faithful and discriminative difficulty proxy.
>
> **Table 1.** Gating entropy statistics across benchmarks. Entropy values exhibit very limited variation (Std $\approx$  0.06), indicating weak discrimination of token difficulty.
>
> | Dataset     | Number of sample | Number of tokens | Gating Entropy per token Mean | Std   |
> |------------:|----------------:|----------------:|------------------------------:|------:|
> | MME        | 33236            | 26199712         | 1.345                         | 0.057 |
> | TextVQA    | 70000            | 58327500         | 1.342                         | 0.061 |
> | GQA        | 176092           | 138376462        | 1.346                         | 0.058 |
>
> We further performed a controlled experiment where gating entropy was used as the difficulty signal in place of log perplexity. As shown in Table 2, this substitution leads to a consistent performance drop across benchmarks, confirming that entropy is not an effective proxy for token difficulty.
>
> **Table 2.** Comparison between log perplexity and gating entropy as difficulty signals. Using entropy results in consistent degradation in accuracy and efficiency.
> | Difficulty Signal     | MME↑ | MMB↑ | TextVQA↑ | SQA↑ |
> |------------------|------:|------:|------:|-------------:|
> | Gating Entropy           | 1352.6  | 65.1  | 55.5  | 69.2       |
> | Log Perplexity (ours)    |**1404.1** | **66.1** | **57.0** | **69.9** |
>
> #### Gradient Norms.
> Gradient-based difficulty estimation measures a token’s influence on parameter updates and is conceptually meaningful, but its dependence on per-token gradient norms makes it computationally infeasible. The required non-aggregated gradients introduce prohibitive costs in two aspects, exacerbated by the scale of MoE models. Loss of computational efficiency: Standard backpropagation is efficient because it aggregates per-token gradients into a single tensor and leverages highly optimized GEMM operations. Computing individual per-token gradients bypasses this mechanism, causing substantial overhead and poor GPU utilization. Prohibitive memory footprint: Per-token gradient computation also requires storing tensors proportional to batch size × total parameters, resulting in memory demands far exceeding typical hardware capacity. Owing to these extreme compute and memory costs, gradient norms are not a practical supervisory signal in our setting.
>
> #### Attention based Metrics.
> Attention statistics such as attention entropy or sparsity describe how tokens aggregate contextual information but do not directly quantify how difficult they are to model. Their relationship to predictive difficulty is indirect and inconsistent across architectures. Consequently, they fail to provide a stable signal for expert allocation.
>
> #### Summary.
> Log perplexity directly measures how confidently the model predicts each token given its context. It is architecture agnostic, interpretable, and computationally lightweight. For these reasons, it serves as a principled and practical proxy for token difficulty in DARE.

---

> ### Author Response · Authors · 2025-11-20
> **Response to reviewer f5dZ (Part 2)**
>
> > Weaknesses 2: Increased Hyperparameter Count
>
> We acknowledge that DARE introduces additional hyperparameters compared with the zero-parameter simplicity of Top-K routing. However, their tuning is straightforward and follows standard deep learning practice.
>
> **(1) Difficulty Predictor ($\alpha$, $lr_{predictor}$).**
> The difficulty predictor is controlled by two hyperparameters, $\alpha$ and $lr_{predictor}$, which jointly regulate the gradient scale of the loss function used in the predictor. These two parameters can be tuned together to maintain stability, and coarse adjustment is sufficient without exhaustive search.
>
> **(2) Online Threshold Adaptation ($\gamma$, $\Pi$).**
> The EMA momentum $\gamma$ serves as a smoothing coefficient similar to momentum in BatchNorm or SGD and shows low sensitivity when set within 0.99–0.999.
> The target distribution $\Pi$ encodes the inductive bias on expert allocation and is the most influential new hyperparameter. As shown in Table 5 of the paper, the long-tailed $\Pi$ significantly outperforms a uniform counterpart, confirming the robustness of this choice.
>
> **Summary.**
> Most new parameters ($\alpha$, lr_predictor, $\gamma$) are standard and insensitive; $\Pi$ is principled, empirically validated, and further detailed in our response to Question 2.
>
>
> > Question 2: Target Distribution $\Pi$ Justification and Ablation
>
> We thank the reviewer for highlighting the importance of analyzing the target distribution  $\Pi$. As noted, Table 5 in the paper already includes an ablation comparing our long-tailed  $\Pi$ with a uniform distribution, showing that allocating more experts to a small number of difficult tokens yields clear performance gains.
>
> Following the reviewer's suggestion, we further trained a model using an inverse (skewed)  $\Pi$ that assigns more experts to difficult tokens and fewer to easy ones. Specifically, we reversed the probability mass from [0.6, 0.3, 0.09, 0.01] to [0.01, 0.09, 0.3, 0.6]. As shown in Table 3, this configuration significantly degrades performance, confirming that the proposed long-tailed  $\Pi$, which prioritizes difficult tokens, is a principled and empirically supported design choice.
>
> **Table 3.** Comparison of different target distributions  $\Pi$.
>
> |  $\Pi$ Type             | Avg-K↓ |MME ↑ | MMB ↑ | TextVQA ↑ | SQA ↑ |
> |-------------------------|------:|------:|------:|------:|------:|
> | Uniform                 | 2.0  |1352.5  | 65.0   | 56.5  | 68.2|
> | Inverse (skewed)        | 3.1 |1378.2  | 65.8  | 55.3  |69.1 |
> | Long-tailed (ours)      | **1.3** |**1404.1**  | **66.1**  | **57.0**  | **69.9**  |
>
> These results demonstrate that the model benefits from allocating more experts to harder tokens, consistent with DARE's difficulty-aware routing objective. We will include this additional analysis in the revised manuscript.

---

> ### Author Response · Authors · 2025-11-27
> **Looking Forward to Your Response**
>
> Dear Reviewer f5dZ,
>
> We greatly appreciate the time you've invested in reviewing our response. Having submitted our rebuttal, we are eager to know if our response has addressed your concern. The discussion phase is nearing its end, but we have not yet received your response. We look forward to hearing from you for any further clarification that you might require.
>
> Best,
>
> Submission 8748 Authors.

---

### Author Response · Authors · 2025-12-03
**Review and Reviewer-Author Discussion Summary**

Dear PCs, SACs, ACs, and Reviewers,

Thank you very much for your valuable contributions to our work. To assist the AC and reviewers in efficiently digesting the rebuttal updates, we provide below a summary of the key points from the reviews and our corresponding responses.

**Strength.** Overall, the reviewers acknowledged the clear motivation, solid empirical results, and the interpretability of our method. Specifically:
* **Motivation:** The paper correctly identifies the limitation of standard Top-K routing and provides a clear motivation for difficulty-aware adaptive routing.
    * All four reviewers recognized this point (**f5dZ**: Strength 1, **VRoW**: Strength 1, **Gk9W**: Strength 1, **bBMU**: Strength 2).
* **Empirical Performance:** The method consistently outperforms various baselines (including static and dynamic routing) across multiple VLM benchmarks and backbones.
    * Three reviewers explicitly highlighted this (**f5dZ**: Strength 2, **VRoW**: Strength 2, **Gk9W**: Strength 2).
* **Clarity and Analysis:** The paper is well-written, providing a "closed loop" between motivation and evidence, with detailed visualizations.
    * Three reviewers highlighted this (**f5dZ**: Strength 3, **VRoW**: Strength 1, **Gk9W**: Strength 3).

**Review and Reviewer-Author Discussion Summary**

**Concerns and Our Addressing.** During the discussion period, we have conducted extensive additional experiments to address the reviewers' concerns. Specifically:

**1. Scalability and Generalization (Number of Experts & NLP Tasks).**
* **(VRoW: Weakness 1, Gk9W: Weakness 3)** Concerns that 4 experts are insufficient to verify scalability.
* **(bBMU: Weakness 4)** Request for validation on language-only tasks.
* **Our Addressing:**
    * We trained a new model using **Qwen2-0.5B with 16 experts**. Results show DARE consistently outperforms Top-K (e.g., +69 points on MME) while reducing activated experts (Avg-K 1.56 vs. 2.0), verifying scalability.
    * We conducted new experiments on **language-only benchmarks** (HellaSwag, ANLI, BoolQ) using Qwen2-0.5B. DARE maintained performance advantages, demonstrating generalization beyond VLMs.

**2. Justification of Design Choices (Difficulty Signal & Target Distribution $\Pi$).**
* **(f5dZ: Weakness 1, Gk9W: Weakness 2)** Why use Log-Perplexity instead of Gating Entropy or Gradient Norms?
* **(f5dZ: Question 2, Gk9W: Weakness 1)** Justification and sensitivity of the target distribution $\Pi$.
* **Our Addressing:**
    * **Signal:** We provided statistical evidence showing that Gating Entropy has extremely low variance (Std $\approx$ 0.06), making it a poor discriminator. We also added a comparative experiment showing that using Entropy significantly degrades performance compared to Log-Perplexity.
    * **$\Pi$:** We performed an ablation study with an "Inverse (skewed)" distribution, which degraded performance, confirming our long-tailed choice is principled. We also experimented with a learnable $\Pi$ (via soft relaxation) as suggested by Reviewer Gk9W, but found it unstable, justifying our fixed-distribution design.

**3. Efficiency Baselines and Overhead.**
* **(VRoW: Weakness 2)** Missing efficiency comparisons (GFLOPs) for dynamic baselines.
* **(bBMU: Weakness 2)** Concerns about the computational overhead of the difficulty predictor.
* **Our Addressing:**
    * We added a comprehensive table comparing **GFLOPs/token** across all methods (MoE++, ReMoE, DynMoE). Results confirm DARE achieves competitive efficiency (64.8 GFLOPs) with reduced Avg-K (1.22).
    * We quantified the predictor overhead, showing it accounts for **<0.01%** of total parameters and FLOPs, confirming it is negligible.

**4. Conceptual Clarifications and Robustness.**
* **(bBMU: Weakness 1)** Distinction from Top-p/DynMoE (Overclaim concerns).
* **(VRoW: Weakness 3)** Request for statistical robustness (error bars).
* **Our Addressing:**
    * We clarified that DARE models *explicit difficulty*, whereas Top-p/DynMoE rely on *uncertainty* or *embedding affinity*, which are theoretically distinct.
    * We reported the mean and standard deviation over **three independent runs**, showing zero variance in performance ($\pm$ 0.0) due to DARE's deterministic routing nature during evaluation.

We believe these additional experiments and clarifications have solidly addressed the concerns raised. We remain committed to incorporating all new results into the final revision and engage in further discussion if needed.

Sincerely,
Authors

---

### Note · Authors · 2026-01-05

I have read and agree with the venue's withdrawal policy on behalf of myself and my co-authors.